# Interaction between subventricular zone microglia and neural stem cells impacts the neurogenic response in a mouse model of cortical ischemic stroke

Suvra Nath[1,2], Jose C. Martínez Santamaría[1,2], Yu-Hsuan Chu [1,2], James S. Choi [3], Pasquale Conforti[1,2], Jia-Di Lin[1,2], Roman Sankowski [4], Lukas Amann [4], Christos Galanis[5], Kexin Wu[1,2], Sachin S. Deshpande[1,2], Andreas Vlachos [5,6,7], Marco Prinz [4,7,8], Jae K. Lee [3] & Christian Schachtrup [1,7] ✉

After a stroke, the neurogenic response from the subventricular zone (SVZ) to repair the brain is limited. Microglia, as an integral part of the distinctive SVZ microenvironment, control neural stem / precursor cell (NSPC) behavior. Here, we show that discrete stroke-associated SVZ microglial clusters negatively impact the innate neurogenic response, and we propose a repository of relevant microglia–NSPC ligand–receptor pairs. After photothrombosis, a mouse model of ischemic stroke, the altered SVZ niche environment leads to immediate activation of microglia in the niche and an abnormal neurogenic response, with cell-cycle arrest of neural stem cells and neuroblast cell death. Pharmacological restoration of the niche environment increases the SVZ-derived neurogenic repair and microglial depletion increases the formation and survival of newborn neuroblasts in the SVZ. Therefore, we propose that altered cross-communication between microglial subclusters and NSPCs regulates the extent of the innate neurogenic repair response in the SVZ after stroke.

The subventricular zone (SVZ) serves as an endogenous stem cell niche for continuous neurogenesis under homeostatic conditions in the adult mammalian brain[1–3]. It thus has the inherent ability to contribute to brain repair processes and to generate new functional neurons when disease damages the central nervous system (CNS). However, for reasons not currently fully understood, the neurogenic response derived from the SVZ after stroke is very limited[4–7]. A fine-tuned cellular and molecular niche environment is instrumental for neural stem/precursor cell (NSPC) maintenance and differentiation[8,9]. Microglia, key immune cells in the CNS[10], are an integral part of stem cell niches[11] and regulate the behavior of NSPCs. Under homeostatic conditions, microglia in the SVZ secrete a distinctive set of cytokines that beneficially affect neurogenesis[12] and express low levels of purine receptors to promote NSPC survival and migration[13]. Pathological

[1]Institute of Anatomy and Cell Biology, Faculty of Medicine, University of Freiburg, Freiburg, Germany. [2]Faculty of Biology, University of Freiburg, Freiburg, Germany. [3]Miami Project to Cure Paralysis, Department of Neurological Surgery, University of Miami School of Medicine, Miami, FL, USA. [4]Institute of Neuropathology, Faculty of Medicine, University of Freiburg, Freiburg, Germany. [5]Department of Neuroanatomy, Institute of Anatomy and Cell Biology, Faculty of Medicine, University of Freiburg, Freiburg, Germany. [6]BrainLinks–BrainTools Center, University of Freiburg, Freiburg, Germany. [7]Center for Basics in NeuroModulation (NeuroModulBasics), Faculty of Medicine, University of Freiburg, Freiburg, Germany. [8]Centre for Biological Signalling Studies (BIOSS) and Centre for Integrative Biological Signalling Studies (CIBSS), University of Freiburg, Freiburg, Germany. ✉e-mail: christian.schachtrup@anat.uni-freiburg.de

states, including stroke, dynamically change the permeability of the SVZ vasculature and drastically alter the finely tuned SVZ stem cell niche environment[14]. However, whether the altered niche environment affects microglia–NSPC interactions and the extent of the neurogenic response after stroke is unknown. In this study, we investigated this interaction in mice subjected to photothrombotic ischemia, a mouse model of cortical ischemic stroke. Here, we show that cortical stroke-evoked environmental changes in the SVZ stem cell niche environment increase NSPC proliferation, arrest the cell cycle of a fraction of type C cells, and induce immediate microglial activation. Consequently, the altered microglial subcluster - NSPC interactions limit the neurogenic repair response after stroke.

## Results

### Impaired neurogenic response and distinct microglial sub-clusters in the SVZ after cortical injury

In a mouse model of cortical ischemic stroke (photothrombotic ischemia, PT), NSPCs in the adult SVZ respond to the distant injury with increased proliferation, formation of doublecortin-positive (DCX +) newborn neuroblasts (type A cells), and redirected migration of these neuroblasts towards the lesion area (Fig. 1a and Supplementary Fig. 1a); however, neurogenic cell replacement in the lesion area is very limited[4–7]. NSPCs in the adult SVZ communicate with the vascular system, the cerebrospinal fluid, and local cells, such as SVZ microglia[8,13], and changes in these interactions may determine NSPC fate and the outcome of repair processes. Upon cortical stroke, the SVZ stem cell niche environment is drastically changed, with increased permeability of the SVZ vasculature and blood-derived fibrinogen deposition[14], and these changes coincide with early SVZ microglial activation (Fig. 1b and Supplementary Fig. 1b). To investigate how the interactions between microglia and NSPCs in the SVZ stem cell niche affect the neurogenic outcome for brain repair under pathological conditions, we performed single-cell RNA sequencing (scRNA-Seq) of selectively isolated SVZ microglia and NSPCs after PT (Fig. 1c). We used a CNS-NSPC-specific inducible nestin mouse line (line $k$[15]) that, upon tamoxifen treatment, produced restricted gene recombination and specific labeling of NSPCs, including robust labeling of DCX + neuroblasts in the SVZ (Supplementary Fig. 1c–e). SVZ-derived CD11b+CD45[low] microglia and YFP + NSPCs were isolated by flow cytometry (Supplementary Fig. 1f), and mCEL-Seq2, a scRNA-Seq protocol that can resolve sparse cell populations[16], was used for sequencing. Although blood-derived CCR2 + monocytes and CD45+ leukocytes were found to barely infiltrate the SVZ stem cell niche after cortical PT (Supplementary Fig. 1g, h), we excluded these cell populations (CD11b + CD45[high]Ly6C/Ly6G +) in the gating strategy to obtain a pure SVZ microglial population. We applied post-sequencing analysis and cluster identification with annotated SVZ microglia and NSPC markers[11,17–27] to define microglia, NSPCs, and a mixed cell population cluster (Supplementary Figs. 1i and 2).

Downstream subclustering analysis of the identified SVZ NSPCs (Supplementary Fig. 1i) based on differentially expressed genes (DEGs)[11,17,19,21,22] identified quiescent type B cells, activated type B cells, type C cells, and neuroblasts (Fig. 1d, e, Supplementary Fig. 3a and Supplementary Data 1). The SVZ NSPC subclusters from uninjured, PT day 1, and PT day 7 mice were organized along a single pseudotime trajectory: cells from the quiescent type B cell cluster were enriched at the vertex, and neuroblasts were enriched at the end point of the trajectory (Fig. 1f). This suggests the SVZ NSPC lineage potential for differentiation towards newborn neurons after a stroke. Further analysis of the proportion of NSPCs revealed an increased number of activated type B cells (Supplementary Fig. 3b), as expected[21], and lineage progression analysis of the NSPC differentiation trajectory revealed the transition through a proliferative type C cell state 1 day after PT (Fig. 1g, black arrows). Immunolabeling for epidermal growth factor receptor (EGFR), which specifically labels activated type B cells[28], and for 5'-ethynyl-2'-deoxyuridine (EdU), which detects

proliferation, in combination with labeling for GFAP and Sox2 (type B cells), confirmed our scRNA-Seq data (Supplementary Fig. 3b) and revealed that cortical injury leads to a robust increase in activated type B cells (Supplementary Fig. 3c) and NSPC proliferation (Supplementary Fig. 3d, e). To promote lineage commitment towards SVZ newborn neuroblasts, NSPCs must re-enter the cell cycle and divide. However, the cell cycle can be blocked or interrupted by changes to the stem cell microenvironment, such as the changes that occur after a stroke, potentially arresting growth and leading to cell death[29]. Mash1 + type C cells increased in number after PT (Supplementary Fig. 3f) and expressed cyclins, such as *Mki67* and *Ccdn2*, indicating their orderly progression towards the G1/S phase of the cell cycle after PT (Supplementary Fig. 3g). However, a fraction of these cells expressed markers linked to the arrest of cell growth, such as *Bcl2* and *Gas6*, indicating cell-cycle arrest (Fig. 1h, i and Supplementary Fig. 3g). Indeed, immunolabeling for DCX in conjunction with ApopTag revealed a drastic increase in apoptotic neuroblasts within the SVZ stem cell niche 7 days after PT (Fig. 1j and Supplementary Fig. 3h). These results show that a distant cortical brain insult leads to neuro-blast cell death in the SVZ stem cell niche, potentially due to drastic changes in the NSPC microenvironment.

Next, we analyzed the molecular composition of SVZ microglia and their functional changes after PT. Downstream subclustering analysis of identified SVZ microglia (Supplementary Fig. 1i) based on DEGs[20,23–27] identified five subclusters (clusters 0–4) (Fig. 1k, Supplementary Fig. 4a and Supplementary Data 1). Cluster 0 was predominant in the uninjured condition (Fig. 1l) and was characterized by the expression of several annotated markers of steady-state patrolling microglia[23], such as *Cx3cr1*, *P2ry12*, and *Gpr34* (cluster 0, Fig. 1m). This homeostasis-associated microglial cluster 0 appeared together with microglial cluster 2 (Fig. 1l), which mostly expressed genes of as-yet unknown function, such as *Gm10800*, *Gm37954*, and *Gm29055* (cluster 2, Fig. 1m). The microglial clusters associated with PT were clusters 1, 3, and 4. Cluster 1 was predominant 1 day after PT (Fig. 1l) and expressed the markers *Srgn*, *Cd300lf*, *Id2*, and *Mt1*, as well as *Socs3* and *Tlr7* (cluster 1, Fig. 1m and Supplementary Fig. 4a), related to microglial activation[30]. Interestingly, 1 day after PT, a unique cluster appeared (cluster 3, Fig. 1l) enriched in genes related to cell-cell adhesion and migration, such as *Elf3*, *Evpl*, *Lad1*, and *Mal2*[31–34] (cluster 3, Fig. 1m). Importantly, 7 days after PT, a microglial cluster appeared that expressed the *Axl*, *Clec7a*, *Apoe*, and *Cd52* genes (cluster 4, Fig. 1l, m), which have been linked to "damage-associated microglia" (DAM)[26]. Microglial clusters 0 and 2 were also present 7 days after PT (Fig. 1l), suggesting that a fraction of the activated microglia had reverted back to a more homeostasis-associated state.

To further delineate the SVZ microglial phenotype, we performed immunohistochemical analysis in microglia-targeting mouse lines. Immunolabeling for Iba1 (ionized calcium-binding adapter molecule) in uninjured *HexB-tdTomato* transgenic mice resulted in ~100% co-labeling of cortical microglia (Supplementary Fig. 4b). By contrast, ~40% of HexB+ cells (red) in the SVZ either expressed low levels of Iba1 or were negative for Iba1 (green) (Supplementary Fig. 4b). In line with this, immunolabeling for Iba1 in *Cx3cr1-CreER[T2]:R26-tdTomato* mice at day 1 and day 7 after PT and in uninjured mice revealed that ~15–20% of RFP + SVZ microglia either expressed low levels of Iba1 or were nega-tive for Iba1 (green) (Supplementary Fig. 4c). Iba1 is a crucial protein for microglial structure, movement, and immune functions. Interest-ingly, ~40% of SVZ-populating Iba1−RFP + microglia expressed CD68 (Supplementary Fig. 4d). Together, these suggest that the dynamic environmental signals in the SVZ neurogenic niche may foster a higher diversity of microglial phenotypes and functionality. A distinct SVZ microglial phenotype that prevents purinergic signaling from trigger-ing premature phagocytosis of migrating neuroblasts under home-ostasis has been described[13]. Indeed, immunolabeling for P2ry12 and C1qb showed low to absent expression in the SVZ microglia of

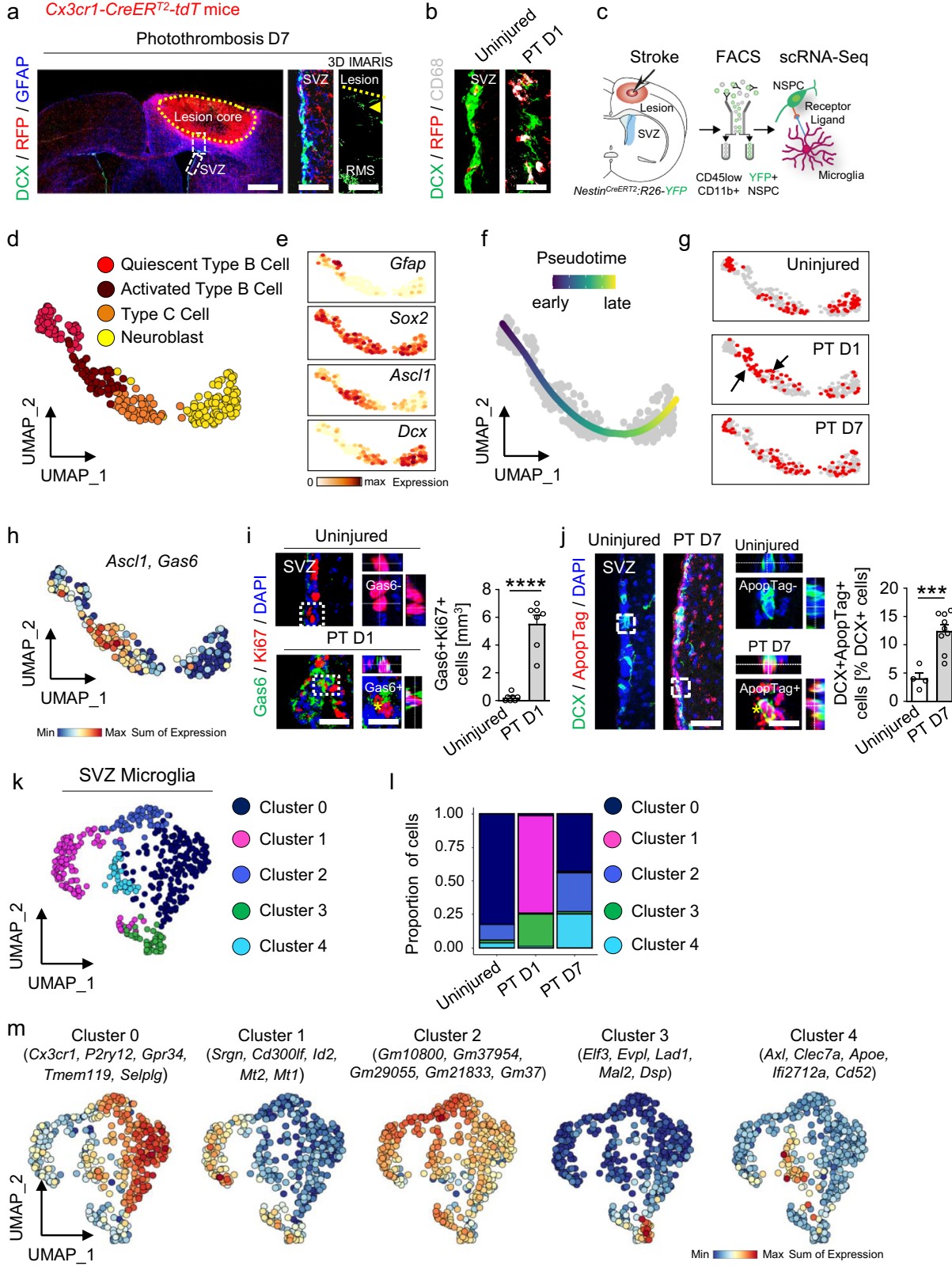

uninjured mice (Supplementary Fig. 4e, f). However, on day 1 after PT, P2ry12 and C1qb expression increased drastically in a fraction of Cx3cr1+ SVZ microglia but returned to homeostatic levels on day 7 after PT (Supplementary Fig. 4e, f). Together, these immunohistochemical data confirm that microglia in the SVZ stem cell niche microenvironment have a heterogeneous molecular composition that shows functional changes after cortical PT.

## Stroke-associated microglia reduce the SVZ neurogenic response

Next, we investigated whether changes in the finely tuned SVZ microenvironment after stroke alter the microglial activation status toward a phenotype that impacts the neurogenic response. Fibrinogen is enriched in the SVZ niche following distant cortical brain injury[14] and induces neurotoxic microglial gene programs in neurodegeneration[35].

**Fig. 1 | SVZ neuroblast cell death and distinct SVZ microglial subclusters after cortical stroke. a** DCX immunostaining (green) in combination with RFP (red) and GFAP (blue) labeling in the SVZ and the lesion penumbra 7 days after PT. White boxes indicate enlarged areas showing the SVZ showing DCX and RFP expression, and IMARIS reconstruction of the lesion penumbra, with few DCX + migratory neuroblasts (arrowhead) ($n = 1$). The lesion core is indicated by a dotted line. Scales (l to r): 750, 100, 260 μm. **b** DCX immunostaining (green) with RFP (red) and CD68 (gray) labeling in the SVZ 1 day after PT, indicating early SVZ microglia activation after cortical stroke ($n = 1$). Scale, 15 μm. **c** scRNA-Seq and analysis of SVZ NSPCs and microglia after PT. **d** UMAP representation of four NSPC clusters from Supplementary Fig. 1i. **e** NSPC cluster-specific gene expression defining their localization along the UMAP of Supplementary Fig. 1i. **f** UMAP visualization of fate-mapped NSPCs captures differentiation from quiescent type B cells to neuroblasts after PT. **g** Pseudotime progression along the NSPC differentiation trajectory after PT. Black arrows indicate type B cell activation and differentiation at day 1 after PT. **h** *Ascl1* +

proliferative type C cells co-express *Gas6* after PT. **i** Immunolabeling for Gas6 (green) and Ki67 (red) in the SVZ 1 day after PT. Dashed boxes indicate magnifications of Gas6 − Ki67 + (top) and Gas6 + Ki67 + cells (bottom, asterisk). Scales, 24 (left), 13 μm (magnified images). Right, quantification ($n = 7$ mice). **j** Immunolabeling for DCX (green) and ApopTag (red) in the SVZ 7 days after PT. Dashed boxes indicate magnifications of ApopTag−DCX + (top) and ApopTag + DCX + cells (bottom, asterisk) of control and 7 days after PT, respectively. Scales, 50 (left), 16 μm (magnified images). Right, quantification ($n = 4$ mice, uninjured; $n = 9$ mice, PT D7). **k** UMAP representation of five distinct microglial clusters obtained after subcluster analysis of the identified microglial cluster in Supplementary Fig. 1i. **l** Quantification of microglial cluster proportions. **m** UMAP plots of expression of core signature genes of SVZ microglial cluster. All graphs show the mean ± SEM. ***$P < 0.001$, ****$P < 0.0001$, unpaired Student's $t$ tests. RMS, rostral migratory stream; SVZ, subventricular zone. Source data are provided as a Source Data file.

Immunolabeling for GFP + microglia in *Cx3cr1-EGFP* mice in combination with immunolabeling of CD31 + blood vessels (red) and fibrinogen (blue) revealed that cortical injury-induced increases in SVZ vasculature permeability and fibrinogen deposition led to a rapid microglial response toward the vasculature and the formation of perivascular clusters (Fig. 2a). Morphometric analysis and 3-D reconstruction of individual Cx3cr1-GFP + SVZ microglia revealed the transformation from an immature microglial morphology in uninjured mice to an ameboid morphology 1 day after PT, to a highly ramified structure with extended branching processes 7 days after PT (Fig. 2b and Supplementary Fig. 5a). To explore the dynamics of interactions between SVZ microglia and NSPCs, we performed time-lapse imaging of acute SVZ explant slices from *Nestin-CreER^T2^-R26-tdTomato:Cx3cr1-EGFP* transgenic mice after cortical PT. In acute explant slices taken on day 7 after PT, GFP + SVZ microglia (green) approached tdT + SVZ NSPCs (red) at a higher speed than in acute explant slices from uninjured mice, and microglia engulfed NSPCs by forming a special phagocytotic pouch (Supplementary Fig. 5b and Supplementary Movie 1). Next, we observed that SVZ stem cell niche microglia phagocytosed DCX + dying neuroblasts (Fig. 2c and Supplementary Fig. 5c, d). Of note, we observed a slightly, but not significantly, increased microglia density 3 days after PT using the *Cx3cr1-CreER^T2^:R26-tdTomato* transgenic mouse line, without profound microglial proliferation or microglial cell death (Supplementary Fig. 5e−g), suggesting a migratory contribution of nearby microglia to the SVZ stem cell niche after PT. Next, to test whether blocking fibrinogen deposition in the SVZ prevented cortical injury-associated changes in the SVZ, we used systemic administration of the pharmacological reagent ancrod to deplete fibrinogen before PT and for the duration of the experiment. Fibrinogen depletion resulted in a ˜ 60% reduction in activated microglia in the SVZ at 10 days post-injury, compared with control mice (Supplementary Fig. 6a). Importantly, fibrinogen depletion revealed a ˜ 2-fold increase in the SVZ-derived DCX+ neurogenic response in the lesion penumbra (Fig. 2d). Moreover, depletion of microglia with the colony-stimulating factor 1 receptor (CSF1R) inhibitor PLX5622 drastically reduced DCX + neuroblast cell death 7 days after PT and increased the overall number of DCX + cells in the SVZ (Fig. 2e and Supplementary Fig. 6b−f). Together, our results indicate that the increased vasculature permeability and SVZ environmental changes after PT induce microglial activation, leading to microglial phagocytosis of neuroblasts and their negative impact on the neurogenic response.

Next, we further characterized the cellular interactions between SVZ microglia and NSPCs that lead to the altered neurogenic response. Microglial cluster expression profiles revealed a significant overlap between genes expressed in the homeostasis-associated microglial cluster 0 and published scRNA-Seq data from uninjured SVZ[11], such as *P2ry12*, *Tmem119*, *Cx3cr1*, and *HexB*; as expected, gene expression in the stroke-associated clusters 1 and 4 did not overlap (Fig. 2f). Gene ontology (GO) term enrichment (adjusted *p*-value < 0.05) analysis

using the DEGs per microglial cluster showed that cluster 0 was involved, as expected, in homeostasis-associated processes, such as actin-filament−based processes, cytoskeletal organization, cell migration, and motility, suggesting a patrolling nature for microglia in the healthy SVZ (Fig. 2g). Stroke-associated cluster 1, which appeared 1 day after PT, was enriched in DEGs associated with responses to stress, oxidative damage, regulation of metabolic processes, and inflammatory responses, and with signaling pathways such as the tumor necrosis factor alpha (TNF), Toll-like-receptor, and signal transducer and activator of transcription 3 (STAT3) pathways, indicating rapid microglial activation after cortical injury. Interestingly, stroke-associated cluster 4, which appeared by day 7 after PT, was linked to key biological processes such as endocytosis and the formation of lytic vacuoles, lysosomes, and vesicles, and also to the major histocompatibility protein complex class 1 (MHC 1), all of which are crucial indicators of microglial phagocytosis of cellular debris (Fig. 2g).

Next, we compared stroke-associated microglial clusters 1 and 4, prevalent in the SVZ stem cell niche after distant cortical injury, with the genes reported in two independent transcriptomic studies of neurodegenerative disease−associated microglia[26,36]. There was a significant overlap between the genes overexpressed in SVZ microglial cluster 4 (7 days after PT) and the genes enriched in neurodegenerative DAM, such as *Apoe*, *Axl*, *B2m*, and *Lgals3bp* (Fig. 2h and Supplementary Data 1), suggesting a potential role for microglial cluster 4 in the clearance of dying neuroblasts within the SVZ stem cell niche after cortical PT. Immunolabeling for ApoE + RFP + microglia and DCX + neuroblasts revealed that SVZ ApoE + microglia was increased 7 days after PT and phagocytosed DCX + neuroblasts (Fig. 2i), suggesting that stroke-associated cluster 4 microglia with DAM-like characteristics phagocytose dying neuroblasts in the SVZ after distant cortical injury.

## SVZ microglia - NSPCs ligand-receptor pair interactions

To gain mechanistic insight into the cellular interactions between SVZ microglia and NSPCs after distant cortical injury, we used CellphoneDB[37] to evaluate interaction scores on the basis of the average expression levels of ligands and receptors in two distinct cell populations (Fig. 3a and Supplementary Data 1). Interestingly, whereas some signaling pathways, such as Igf2, Kitl, and Gnai2, were only present in uninjured mice, other signaling pathways were attenuated 1 day after PT and upregulated 7 days after PT (e.g., Igfbp4, Npnt, Ly86, Sema4d, Calr), suggesting a drastic change in microglia−NSPC interaction 1 day after PT, when microglia become activated because of changes in the SVZ stem cell niche environment (Fig. 2a). Strikingly, the highest interaction score between microglia and NSPCs was detected for the ApoE−Lrp8 pathway, which was uniquely and significantly upregulated 7 days after PT and which correlated with the appearance of stroke-associated microglial cluster 4 (Fig. 3a and Supplementary Data 1). Indeed, chord diagrams revealed a high interaction score between microglia cluster 4 and NSPCs seven days after PT

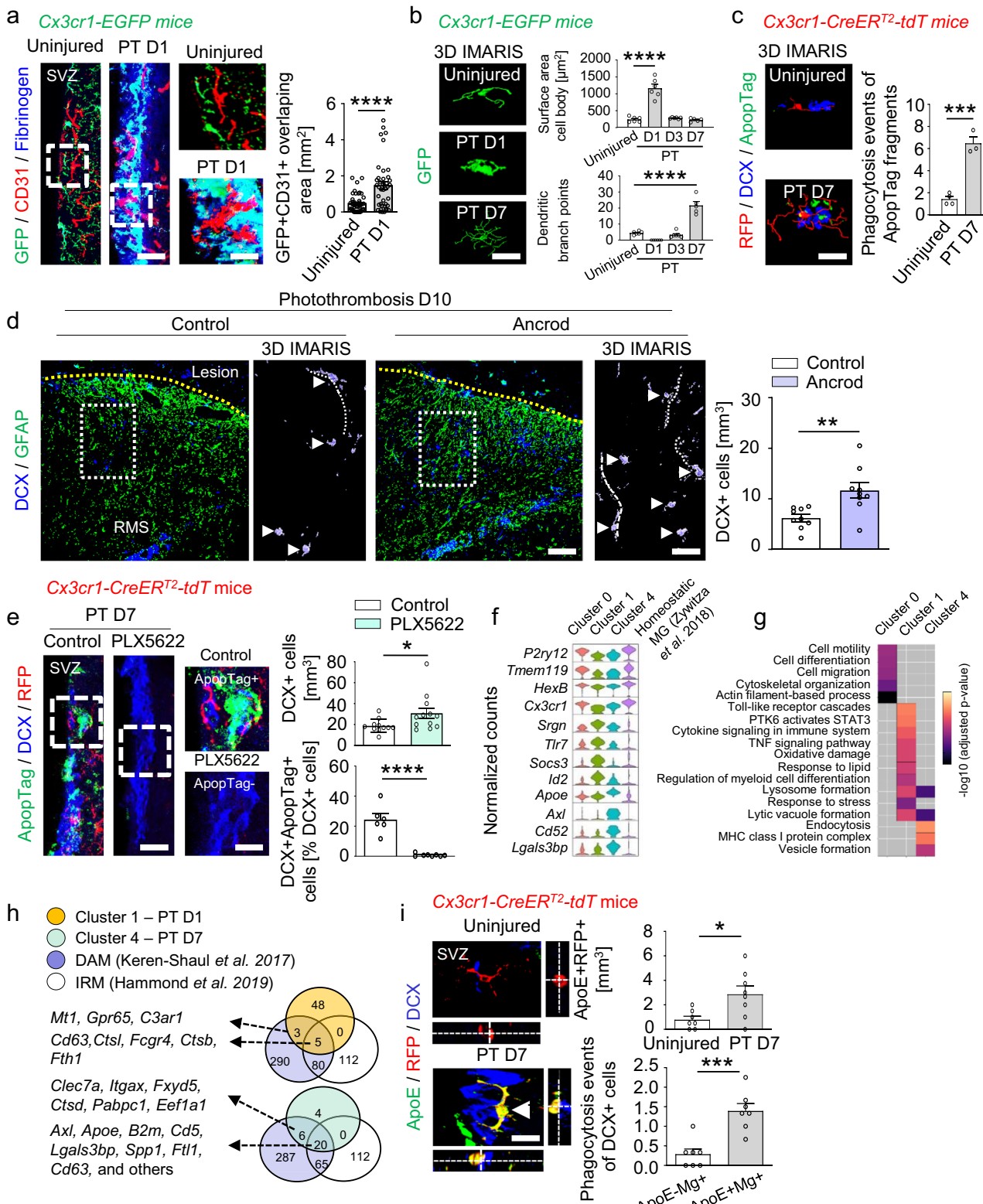

(Fig. 3b), and immunolabeling for ApoE and Lrp8 showed expression in SVZ microglia and NSPCs, respectively (Fig. 2i and Supplementary Fig. 7a). Next, to pinpoint microglia–NSPC interaction, we performed in situ hybridization of the ligand-receptor pair ApoE–Lrp8 in combination with immunohistochemistry. *Apoe* and *Lrp8* mRNAs were significantly upregulated in GFP + microglia (green) and Nestin-TdTomato + NSPCs (red), respectively, 7 days after PT, compared with uninjured controls (Fig. 3c). Next, we analyzed the proximity between

ApoE + microglia and Lrp8 + NSPCs (Fig. 3d). In situ hybridization for *Apoe* and *Lrp8* mRNA revealed *Apoe* + cells were in close proximity to *Lrp8* + cells, suggesting interaction between ApoE + microglia and Lrp8 + NSPCs (Supplementary Fig. 7b). Cell proximity analysis of 3-D IMARIS processed images revealed close contact between Lrp8 + DCX + neuroblasts and RFP + microglia and between ApoE + RFP + microglia and DCX + neuroblasts in the SVZ of mice 7 days after PT (Fig. 3e, f). Importantly, alleviating the cortical injury-induced SVZ

**Fig. 2 | Stroke-associated microglia phagocytose dying neuroblasts and reduce the SVZ neurogenic response. a** Immunolabeling for GFP (microglia, green), CD31 (blood vessels, red), and fibrinogen (blue) 1 day after PT. Scales, 18 (left), 10 μm (magnified images). Quantification of overlap between GFP + microglia and CD31 + blood vessels 1 day after PT ($n = 3$ mice; 46 cells, uninjured; 40 cells, PT D1). **b** Reconstructions of microglia (green) and IMARIS quantification of GFP + microglia 1, 3, and 7 days after PT ($n = 6$ mice, uninjured, PT D1, PT D3; $n = 5$ mice, PT D7). Scale, 7 μm. **c** IMARIS reconstruction of DCX + ApopTag + dying neuroblasts engulfed by RFP + microglia (red) 7 days after PT (bottom). Scale, 7 μm. Quantification of ApopTag + fragments in RFP⁺ microglia in the SVZ ($n = 4$ mice, uninjured; $n = 3$ mice, PT D7). **d** Immunolabeling for DCX (blue) and GFAP (green) in the lesion penumbra 10 days after PT. Dotted rectangles indicate magnified images. Arrowheads indicate individual DCX + cells, and dotted lines indicate DCX + processes. Scales, 130 (left), 63 μm, (magnified images). Right, quantification 10 days after PT ($n = 9$ mice). **e** Immunolabeling for ApopTag (green), DCX (blue), and RFP (red) in the SVZ 7 days after PT in PLX5622-fed mice. Dashed rectangles indicate magnified images showing ApopTag + DCX + (top) and ApopTag–DCX + cells (bottom). Scales, 38 (left), 7 μm (magnified images). Quantification of DCX + cells (top) and

percentage of ApopTag + DCX + cells (bottom) 7 days after PT in PLX5622-fed mice (top: $n = 11$, control; $n = 12$, PLX5622; bottom: $n = 6$, control, $n = 8$, PLX5622). **f** Violin plots of markers enriched in stroke-associated microglial clusters 1 and 4, compared with cluster 0 and ref.[11]. **g** Gene ontology term enrichment analysis (adjusted $p$-value < 0.05) using the same clusters. Gray boxes indicate no enrichment. **h** Venn diagram showing the overlap between genes of neurodegenerative disease–associated microglia (purple, disease-associated microglia (DAM)[26]; and white, injury-responsive microglia (IRM)[36]) and DEGs 1 (orange) and 7 days (green) after PT. **i** Immunolabeling for ApoE (green), RFP (red), and DCX (blue), 7 days after PT. Arrowhead indicates ApoE + RFP + microglia phagocytosing a DCX + neuroblast. Scale, 14 μm. Quantification in uninjured mice and 7 days after PT (top) ($n = 7$ mice, uninjured; $n = 8$ mice, PT D7) and of DCX + cells engulfed by microglia (Mg, bottom) ($n = 7$ mice) in the SVZ 7 days after PT. All plots show mean ± SEM. *$P < 0.05$, **$P < 0.01$, ***$P < 0.001$, ****$P < 0.0001$, one-way ANOVAs with Bonferroni corrections for multiple comparisons (**b**) and unpaired Student's $t$ tests (**a**, **c–e**, **i**). RMS, rostral migratory stream; SVZ, subventricular zone. Source data are provided as a Source Data file.

environmental changes by fibrinogen-depletion with ancrod reduced the ApoE + SVZ microglia (Fig. 3g), in line with the increased SVZ-derived neurogenic response in ancrod-treated animals (Fig. 2d), while the proportion of Lrp8 + DCX + neuroblasts remained unaltered (Supplementary Fig. 7c). Overall, these results reveal that cortical injury-induced alterations of the neural stem cell niche environment alter the interactions between SVZ microglia and NSPCs, limiting the neurogenic response after PT.

## Discussion

Understanding how an altered SVZ niche environment affects the interaction between microglia and NSPCs and whether this impacts the extent of the neurogenic response after stroke is a fundamental and unresolved question. Our results suggest that upon cortical injury, increased permeability of the SVZ vasculature and drastic changes to the niche environment (e.g., fibrinogen deposition) alter the microglia–NSPC interaction, with subsequent effects on the neurogenic response. We suggest the following working model in CNS disease: (i) A cortical brain injury-induced increase in SVZ vasculature permeability allows deposition of vasculature-derived proteins, including fibrinogen, altering the microenvironment of SVZ stem cells (Fig. 3h). (ii) The altered SVZ niche environment increases NSPC proliferation and cell-cycle arrest of a fraction of type C cells and induces immediate microglial activation. (iii) Microglia phagocytose apoptotic newborn neuroblasts in the SVZ through the predicted ligand-receptor pair ApoE–Lrp8, resulting in limited neurogenic cell replacement in the cortical lesion area.

Stroke induces only a small number of SVZ-derived newborn neurons: the majority of cells are undifferentiated precursors and newborn astrocytes, which contribute to repair processes in different ways[7]. We previously found that distal cortical injury leads to massive deposition of the blood coagulation factor fibrinogen into the SVZ niche environment and drives NSPC differentiation into astrocytes via BMP signaling[14]. Although the SVZ generates newborn neurons that can contribute to neuronal replacement and functional recovery after stroke, the inhibitory environment around the lesion impedes innate regenerative efforts[38]. Here, we demonstrate that, after cortical injury, the increased permeability of the SVZ vasculature and alterations to the niche microenvironment drastically impact the neurogenic response in the SVZ.

Microglia in the adult SVZ boost the survival and migration of neuronal progenitors under homeostatic condition[13], and stroke (middle cerebral artery occlusion, MCAO) induces a long-lasting accumulation of a proneurogenic microglia phenotype in the SVZ[39]. Our study revealed that this homeostatic, pro-neurogenic SVZ microglial phenotype switched immediately upon cortical stroke, negatively

impacting the neurogenic response for brain repair. Our data suggest that the altered SVZ niche environment evoked by damage to the brain induces a detrimental SVZ microglial state, reducing SVZ neuroblast survival and negatively impacting the innate neuroregenerative processes. Indeed, restoration of the niche environment decreased microglia activation and increased the SVZ-derived neurogenic repair, and microglial depletion increased the formation and survival of newborn neuroblasts in the SVZ (this study). Together, our previous and current work shows that the altered SVZ niche environment after cortical injury induces NSPC-derived astrogenesis[14] and alters the interaction between SVZ microglia and NSPCs (this study), limiting the innate neurogenic response. Future experiments should address whether different CNS diseases impact the SVZ niche cytoarchitecture and impede SVZ-derived repair processes.

Microglial heterogeneity occurs in a region- and context-dependent manner, owing to their high responsivity to diverse environmental conditions[23]. Nevertheless, the heterogeneity of microglia in the neurogenic niches and the biological implications of this heterogeneity during brain injury and reparative processes remain unclear. We report here heterogeneous microglia activation states in the SVZ stem cell niche under homeostatic conditions and distinct stroke-associated microglial states after PT, and we propose that these states differ in terms of their interaction with NSPCs. Our data reveal the specific transcriptional status of microglial subclusters in different regions and in the context of CNS damage[20,23,25,40]. Our scRNA-Seq results suggest that SVZ microglia display distinct states that change chronologically after stroke, potentially because of the unique environment in the stem cell niche that is not present in other brain regions. We identified stroke-associated microglial states (clusters 1 and 4) with divergent features in the SVZ after PT. Subcluster 1 was linked to pro-inflammatory signaling and predicted TNF cross-communication between NSPCs and microglia, potentially arresting the NSPC cell cycle and inducing neuroblast cell death. Interestingly, adult NSPCs reveal dual effects of TNF-α in the regulation of entry to and exit from quiescence[41]. The potential contribution of TNF interaction between microglia and NSPCs on the cell cycle and inducing neuroblast cell death should be studied in detail in the future. Cluster 4 had characteristics associated with DAM, expressed lipid metabolism genes, such as ApoE, and potentially interacted with NSPCs through the low-density lipoprotein (LDL) receptor-related family of proteins (LRPs). DAM has elevated expression of genes involved in lipid metabolism and phagocytosis, corresponding to the need for plaque clearance in Alzheimer's disease[26], but our study revealed increased neuroblast phagocytosis by microglia in stroke-associated SVZ cluster 4 and predicted cross-communication with SVZ NSPCs via the ligand-receptor pair ApoE–Lrp8. Phagocytosis of apoptotic neurons induces

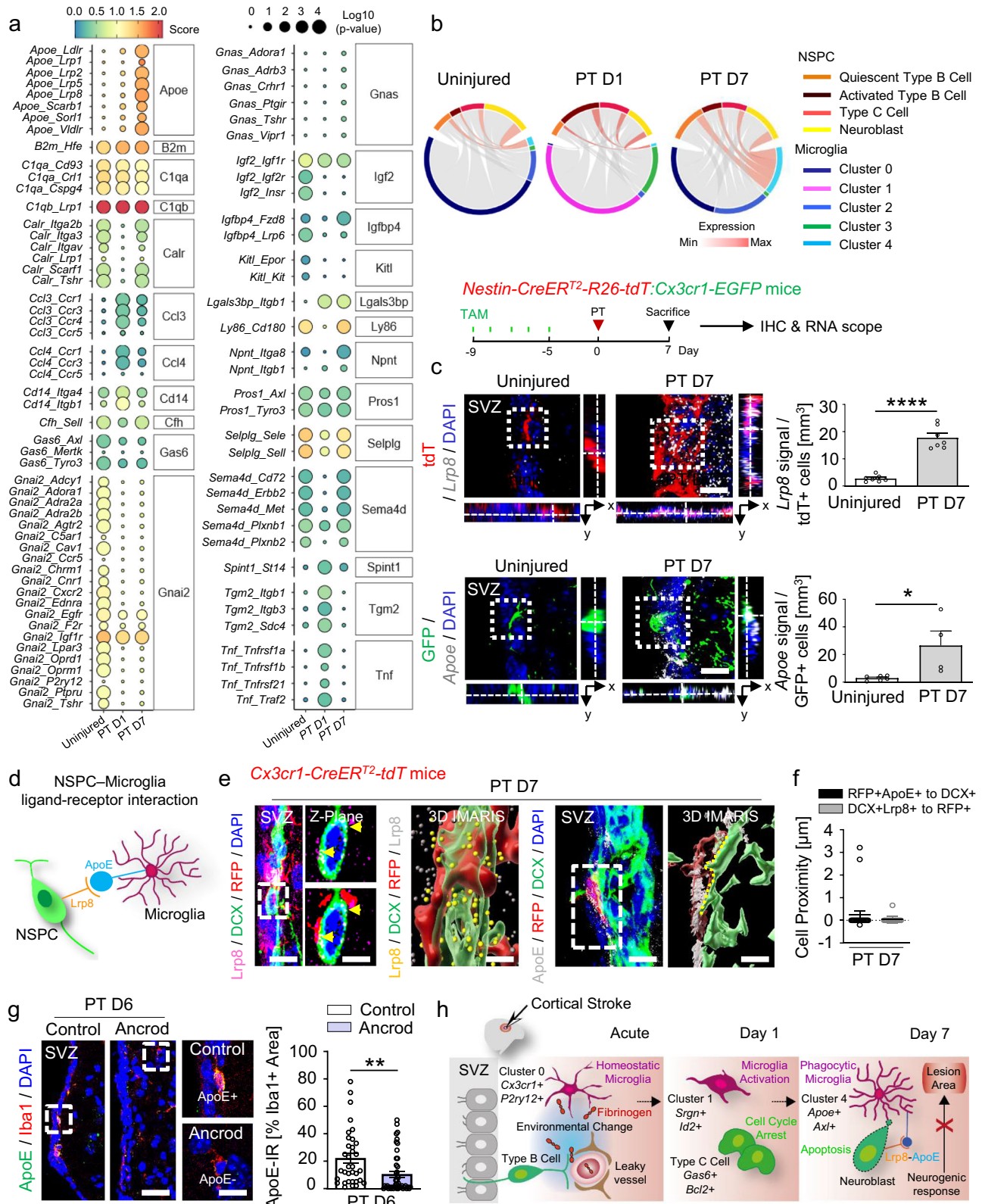

ApoE signaling in microglia and their subsequent suppression of a homeostatic phenotype[27,42]. Our data suggest a role for NSPC-expressed LRPs in ApoE-induced signaling in microglia. The diverse roles of LDL receptor family members, including Lrp8, in regulating NSPC differentiation and functionality are just emerging[43,44].

In general, leveraging SVZ neuroregenerative processes could provide targets for stroke recovery. An emerging concept is the reciprocal interaction between inflammatory cells and NSPCs in CNS disease[45]. Our analysis sheds light on cross-communication between microglia and NSPCs within the SVZ under homeostatic conditions and after stroke. As NSPC–microglia ligand-receptor interactions in the healthy CNS and during disease are still poorly defined, our repository of ligand-receptor pairs will help elucidate the biology of NSPCs and microglia in future studies. It will be important to determine how

**Fig. 3 | Ligand-receptor pair interactions between NSPCs and microglia after cortical stroke. a** Dot plot of interaction scores for ligand-receptor pairs and operating signaling pathways 1 and 7 days after PT. **b** Chord plot of ApoE-Lrp8 ligand-receptor interactions across clusters and NSPC subpopulations within the consensus atlas, based on co-expression 1 and 7 days after PT. **c** In situ hybridization for *Lrp8* (gray) with immunolabeling for RFP + NSPCs (red) in the SVZ 7 days after PT. Scale bar, 12 μm. Quantification of *Lrp8* + RFP + cells 7 days after PT (*n* = 7 mice). In situ hybridization for *Apoe* (gray) with immunolabeling for GFP + microglia (green) in the SVZ 7 days after PT. Scale, 12 μm. Quantification of *Apoe* + GFP + cells (*n* = 6, uninjured; *n* = 4, PT D7). **d** Schematic of the NSPC-microglia Lrp8-ApoE cross-communication. **e** Left images: Immunolabeling for Lrp8, DCX, and RFP in the SVZ 7 days after PT. Higher magnification of dashed square showing Z-plane with (bottom) and without (top) RFP illustrating the location of Lrp8 (arrowheads) on DCX + cells in close contact with RFP + cells. IMARIS reconstruction of Lrp8, DCX, and RFP. DCX + cell surface expression of LRP8 (yellow dots) in contact with RFP + cells. Right pair of images: Immunolabeling for ApoE, RFP, and DCX 7 days after PT. IMARIS reconstruction of the ApoE + RFP + cell in the dashed rectangle in contact with a DCX + cell. The area of contact between the RFP + and DCX + cells is indicated by the yellow dashed line. Scales, 12 μm (SVZ, Lrp8); 6 μm, magnified images; 3 μm, 3-D IMARIS; 8 μm (SVZ,

ApoE); 6 μm, 3-D IMARIS. **f** Quantification of the smallest distance between RFP + ApoE + cells and DCX + cells and between DCX + Lrp8 + cells and RFP + cells 7 days after PT (*n* = 3 mice; 23 pairs of cells, RFP + ApoE + cells to DCX + cells; 18 pairs of cells, DCX + Lrp8 + to RFP + cells). **g** Immunolabeling for ApoE and Iba1 in fibrinogen-depleted mice 6 days after PT. Dashed boxes indicate the magnification of ApoE + Iba1 + (top, control) and ApoE–Iba1 + cells (bottom, ancrod) 6 days after PT. Scales, 28 μm, left; 10 μm, magnified images. Right, quantification of the ApoE immunoreactivity (IR) in Iba1 + cells in fibrinogen-depleted mice compared with control-treated mice 6 days after PT (*n* = 6 mice, control group (34 cells); *n* = 5 mice, ancrod group (44 cells)). Plots show mean ± SEM. *$P < 0.05$, **$P < 0.01$, ****$P < 0.0001$, unpaired Student's *t* tests. **h** In the healthy brain, NSPCs of the SVZ generate mobile DCX + neuroblasts that migrate through the RMS to the olfactory bulb to become newborn neurons. Cortical injury (cortical stroke, acute) results in increased permeability of the SVZ vasculature and a drastic change in the SVZ stem cell niche environment (e.g., fibrinogen deposition). This induces increased NSPC proliferation, cell-cycle arrest of type C cells, and immediate microglial activation (cortical stroke, day 1 after PT). Microglia phagocytose apoptotic newborn neuroblasts through the predicted ligand-receptor pair ApoE–Lrp8, resulting in limited neurogenic cell replacement in the cortical lesion area (cortical stroke, day 7 after PT). SVZ, subventricular zone. Source data are provided as a Source Data file.

different SVZ microglial activation states impact innate SVZ-derived regenerative efforts. Strategies to modulate SVZ NSPC–microglia cross-communication may be a promising therapeutic approach to promote the endogenous SVZ neurogenic response for brain repair after stroke, and potentially in other CNS diseases as well.

## Methods
### Animals
C57BL/6 N mice (Charles River) and transgenic mice (C57BL/6 N background) were used. For the analysis of SVZ NSPCs, *Nestin-CreER^T2* mice[15] were crossed with *R26-yfp* mice[46], resulting in *Nestin-CreER^T2:R26-yfp* mice. For the analysis of microglia, *Cx3cr1-CreER^T2* mice[47] were crossed with *R26-tdTomato* mice to generate *Cx3cr1-CreER^T2:R26-tdTomato* mice. To visualize microglia and monocytes, *Cx3cr1-EGFP*[48], *HexB-tdTomato*[49] and *Ccr2-RFP*[50] transgenic mouse lines were used. For in vitro acute slice imaging, *Nestin-CreER^T2:R26-tdTomato* mice were crossed with *Cx3cr1-EGFP* mice, resulting in *Nestin-CreER^T2-R26-tdTomato:Cx3cr1-EGFP* mice. Adult mice of both genders were used. Animals were housed under Institutional Animal Care and Use Committee guidelines in a temperature and humidity-controlled facility with a 12 h light–12 h dark cycle and *ad libitum* feeding. All animal experiments were approved by the Federal Ministry for Nature, Environment, and Consumer Protection of the state of Baden-Württemberg (G16/110, G21/56) and were performed in accordance with the respective national, federal, and institutional regulations.

### Tamoxifen treatment
For the induction of Cre recombinase in *Nestin-CreER^T2:R26-yfp* (8–12 weeks old), *Nestin-CreER^T2-R26-tdTomato:Cx3cr1-EGFP* (8–12 weeks old), and *Cx3cr1-CreER^T2:R26-tdTomato* (6 weeks old) mouse lines, mice were injected subcutaneously with a daily dose of tamoxifen (TAM, Sigma Aldrich, T6548) (180 mg/kg body weight/d, dissolved in corn oil (Sigma Aldrich)) for five consecutive days. PT was performed 5–14 days after the last injection.

### Ancrod treatment
To investigate the fibrinogen-induced phagocytosis of DCX + neuroblasts by SVZ niche microglia after PT, C57BL/6 mice were depleted of fibrinogen with ancrod (NIBSC, 74/581)[14,51]. Briefly, mice received 2.4 U ancrod, 2.0 mg per kg body weight, or saline solution (control), per day via osmotic minipumps (Alzet, Model: 2002, 0.5 μl/h) implanted subcutaneously in their back 2 d before stroke induction. Ancrod was secreted for the duration of the experiment until the mice were perfused.

### Stroke model
Photothrombotic ischemia (PT) was used to induce stroke in the cortex of adult mice as described previously[14,52]. Briefly, 20 min after injection of the photosensitive dye Rose Bengal (Sigma–Aldrich, 330000; 10 μl/g body weight, intraperitoneal), a cold light illuminator was positioned stereotaxically to deliver light (150 W) to the cortex (AP, 0 mm; ML, − 2.4 mm, according to Paxinos and Watson). The region of interest (4 mm diameter) was illuminated for 4 min, and after the light exposure ended, the wound was sutured.

### SVZ compartment microdissection and flow cytometry
Mice were anesthetized with ketamine and xylazine and transcardially perfused with PBS. Brains were immediately harvested and placed in the dissection solution (0.8% glucose and 15 mM HEPES in HBSS) on ice. The lateral SVZ from the hemisphere ipsilateral to the cortical PT site was microdissected according to published protocols[53]. The SVZ tissue was digested with papain (Worthington, 1200 units per 5 mice) in PIPES solution [120 mM NaCl, 5 mM KCl, 50 mM PIPES (Sigma-Aldrich), 0.6% glucose, 1 × Pen/Strep in water, pH adjusted to 7.6] for 10 min at 37 °C, and mechanically dissociated into single cells with fire-polished glass Pasteur pipettes after adding ovomucoid (Worthington, 0.7 mg per 5 mice) and DNase I (Invitrogen, 1000 units per 5 mice) as described[54]. Myelin debris was removed by centrifuging in 22% Percoll (Sigma-Aldrich) for 10 min at 4 °C without brakes.

For single-cell RNA sequencing, cells were incubated in FACS buffer (1% BSA and 1% glucose in HBSS) for 30 min on ice with the following antibodies: anti-CD133 (biotinylated, 1:300, eBioscience), anti-CD11b (Clone: M1/70, Brilliant Violet 605, 1:300, Biolegend), anti-CD45 (Clone: 30-F11, APC-Cy7, 1:200, Invitrogen), anti-Ly6C (Clone: AL-21, anti-Alexa Fluor 700, 1:200, BD Bioscience), anti-CD3e (Clone: eBio500A2, PE, 1:300, eBioscience), anti-CD19 (Clone: MB19-1, PE, 1:200, eBioscience), anti-Ly6G (Clone:1A8, PE, 1:200, BD Bioscience), Fc Block (Clone: 2.4G2, 1:250, BD Bioscience), Live/Dead (DAPI, 1:10000). FACS sorting was performed on a MoFlo Astrios (Beckman Colter) using a 100-μm nozzle at 13.1 psi pressure. Cell gates were defined as follows: microglia CD11b + CD45^low and NSPC YFP + cells were sorted on a MoFlo XDP Cell Sorter (Beckman Colter, USA) into 384-well plates containing prepared lysis buffer and processed with the CEL-Seq2 modified protocol[55].

### Single-cell RNA sequencing
SVZ NSPCs and microglia were obtained as detailed above. Single-cell RNA sequencing was performed using the mCEL-Seq2 protocol, an automated and miniaturized version of CEL-Seq2 on a mosquito

nanoliter-scale liquid-handling robot[16,56]. Eight libraries with 192 cells each were sequenced per lane on an Illumina HiSeq 3000 sequencing system (pair-end multiplexing run) at a depth of ~130,000–200,000 reads per cell[16]. For quantification of transcript abundance, single cells were sorted in 384-well plates containing RT (reverse transcription) primers (anchored polyT primers having a 6-bp cell barcode, 6-bp unique molecular identifiers (UMIs), part of a 5′ Illumina adapter, and a T7 promoter), dNTPs, Triton X-100 and Vapor-Lock (Qiagen). Prior to sorting, the MoFlo XDP Cell Sorter was calibrated to dispense single cells in the center of each well, and the machine was run using trigger pulse width to exclude doublets. The 384-well plates containing the sorted cells were centrifuged at maximum speed and stored at −80 °C until library preparation[55]. Libraries were sequenced on a HiSeq3000 Illumina sequencing machine, and the de-multiplexing of the raw data was performed by running bcl2fastq (version 2.17.1.14.).

### Single-cell RNA sequencing data analysis

Three 384-well plates (1152 cells in total) were processed using the mCEL-Seq2 protocol, which uses a 6-bp nucleotide sequence as UMI and another 6-bp nucleotide sequence for barcoding. For quality control, the median-absolute deviation of the total UMIs per cell was computed. Cells with total UMI counts greater than three median-absolute deviations from the median were considered putative doublets and discarded. Cells with greater than 1000 UMIs (i.e., a total of at least 1000 gene counts) were retained, and all others were discarded. For each cell, the percentage of UMIs mapping to mitochondrial gene products (e.g., mt-Co2) was calculated as an indicator of cell quality. Cells with greater than 25% of their library mapping to mitochondrial genes were discarded.

Downstream analysis was performed using the Seurat R package (v4.0.2,[57]). Unless otherwise noted, default parameter values were used. First, count data were normalized to library size and transformed using SCTransform. Next, PCA was performed using the top 2000 variable genes, and the top 11 principal components were selected for linear dimensional reduction. FindNeighbors () was run using 11 PCs, and FindClusters () was run with resolution = 2.0. Cells and clustering results were projected onto 2-D space using the non-linear dimensional-reduction technique UMAP (uniform manifold approximation and projection).

To identify the major cell types, a combination of visual inspection of canonical marker genes and globally distinguishing differential gene expression tests were performed. For all differential expression tests, a Wilcox Rank Sum test was performed using all genes, with a Bonferroni correction for multiple testing, as implemented in Seurat's FindMarkers () function. Next, the SingleR R package (v1.6.1)[58] was used to perform automated cell-type annotation. SVZ reference data for the adult mouse brain were taken from Zywitza et al.[11]. Automated cell-type annotations were largely in agreement with manual cell-type labels. In addition, a combined cluster analysis between data from the current study and Zywitza's study provided further confidence in the accuracy of cell-type labeling. Importantly, we did not observe a separation between studies due to technical artifacts such as mitochondrial gene percentage or sequencing platform (Supplementary Fig. 2). To better understand cell-type-specific responses after PT, a subclustering analysis was performed on NSPC and microglia separately. For each cell type, variable gene selection, SCTransform, PCA, and UMAP reduction were repeated. For NSPCs, the top eight PCs were used, and resolution = 0.8 was selected in FindClusters (). For microglia, the top seven PCs were used, and resolution = 0.4 was selected in FindClusters (). For the trajectory analysis, the Slingshot R package was used[59]. Briefly, new UMAP coordinate embeddings for each cell type were used as input for Slingshot. The resulting pseudotime values and lineages were overlaid on UMAPs.

To characterize the microglial subclusters, lists of differentially expressed genes (DEGs) were curated from various scRNA-seq studies

on microglia[20,23–27,36]. DEGs were computed between the injury- or damage-associated microglia and stroke-associated microglia for each study, respectively. Only genes with a log2 fold-change threshold of 0.25 and adjusted p-value less than 0.001 were retained, and sets of DEGs that were shared between studies or unique were tabulated. Venn diagrams of overlapping DEGs were generated using the Eulerr R package (https://cran.r-project.org/web/packages/eulerr/index.html). For enrichment analysis, differentially expressed genes with adjusted $p < 0.05$ of microglia clusters 0, 1, and 4, were subjected to g:Profiler, a public web server (https://biit.cs.ut.ee/gprofiler/gost). The queried list in the Gene Ontology (GO) Biological Process was then plotted using the ggplot2 R package.

### Ligand–receptor analysis

To investigate potential signaling pathways between NSPCs and microglia, the CellPhoneDB method[37] was adapted to perform an unbiased ligand-receptor co-expression screen. Briefly, a reference list of known human ligand-receptor pairs was pulled[60] and was converted into mouse orthologs using the biomaRt R package (v2.48.0)[61]. The ligand-receptor score was defined as the mean of the average log-normalized expression of the receptor gene by one cell type and the average log-normalized expression of the ligand gene by another cell type. To identify enriched ligand-receptor pairs, a permutation test was performed with 1000 random shuffles of cell type and time-point classification. This generated a null distribution of interaction scores for each ligand-receptor pair. Lastly, the interaction scores of the actual labels (ligand cell A, receptor cell B, time-point) were compared to the null distribution, and p-values were calculated as the proportion of null scores that were equal to or greater than the actual interaction score.

### In situ hybridization and RNA scope

Fixed frozen sections mounted on super frost slides were prepared as described above. In situ hybridization was performed using the RNA scope Multiplex Fluorescent Kit v2 (Advanced Cell Diagnostics Cat #: 323100). After washing with PBS, slides were incubated at 60 °C for 30 min (all incubations were performed in an Advanced Cell Diagnostics Hyb-EZ Oven). After 15 min post-fixing in 4% PFA/PBS at 4 °C, slides were dehydrated in a graded ethanol series and incubated in 1 × Target Retrieval Solution at 95 °C for 5 min. Slides were washed with distilled water and rinsed with 100% EtOH. Slides were left to air-dry, followed by incubation in RNAscope Protease III solution at 40 °C for 30 min. After incubation, slides were washed with distilled water and incubated in probe solution for 2 h at 40 °C. The following probes were purchased from Advanced Cell Diagnostics (ACD): Mm-Apoe-C2 (Cat #:313271-C2, 1:50), Mm-Apoe (Cat #:313271, working dilution), Mm-Lrp8-C2 (Cat #:416801-C2, 1:50). After probe hybridization, each amplifier was individually incubated for either 15 or 30 min at 40 °C, according to the manufacturer's protocol. After each individual fluorophore incubation, slides were washed with RNAscope 1 × Wash Buffer. After in situ hybridization, slides were washed with PBS, and immunohistochemistry was performed with appropriate primary antibodies as described above, followed by nuclear counterstaining with DAPI.

### Microglia depletion

For pharmacological ablation of brain microglia, mice were fed PLX5622-formulated AIN-76A diet (1.2 g PLX5622 per kilogram of diet, Plexxikon) ad libitum[62], with normal AIN-76A diet (Plexxikon) as control. Mice were fed with the PLX5622 diet for 12 days. PT was performed 5 days after the start of the PLX5622 diet, and mice were sacrificed 7 days after PT.

### EdU labeling

To label proliferating SVZ microglia and NSPCs, C57BL/6 mice were intraperitoneally injected with EdU (50 mg/kg body weight,

Invitrogen) either immediately after PT and then at 6, 12, and 18 h after PT (for PT D1) or for 3 consecutive days before sacrifice (for PT D3 and PT D7). Mice were sacrificed 4 hours after the last injection.

## Acute slice imaging and image processing

*Nestin-CreER^T2-R26-tdTomato:Cx3cr1-EGFP* mice were decapitated under ketamine/xylazine anesthesia; the brain was quickly removed and placed in Petri dishes filled with ice cold artificial cerebrospinal fluid (ACSF) composed of 126 mM NaCl, 2.5 mM KCl, 26 mM NaHCO$_3$, 1.25 mM NaH$_2$PO$_4$, 2 mM CaCl$_2$, 2 mM MgCl$_2$, and 10 mM glucose and was saturated with 95% O$_2$/5% CO$_2$. The brain was cut coronally into 350-µm-thick slices using a vibratome (VT1200S, Leica) at a speed of 0.06 mm/s and vibration amplitude of 1.5 mm in ice-cold ACSF. The brain slices were allowed to recover in ACSF for 30 min at 32 °C and then adapted to RT. Slices were imaged with a Zeiss laser-scanning confocal microscope (LSM 800, Zeiss) and perfused continuously with bath solution (ASCF saturated with 95% O$_2$/5% CO$_2$) at a rate of 3 ml/min, maintained at 35 °C. Confocal images were collected as z-stacks at 2-minute intervals for 40–45 min (hyperstacks). Image analysis was performed with FIJI software (https://imagej.net/Fiji/Downloads). The hyperstacks were corrected for drift using the Correct 3D Drift and PoorMan3Dred plugins. Phagocytotic events were counted manually using XY and XZ projections. Microglial density was measured using the Cell Counter plugin, and microglial trajectories were calculated with the MTrackJ plugin.

## Histology and immunohistochemistry

Mice were transcardially perfused with ice-cold saline, followed by 4% PFA in phosphate buffer under ketamine and xylazine anesthesia, and brain samples were removed, cryoprotected, embedded in OCT (Tissue-Tek), and frozen on dry ice. Brain samples were cut into 30- or 14-µm coronal sections and immunohistochemistry was performed on these sections as described[63]. For EdU detection, we used the Click-iT Plus EdU Imaging kit (Fisher Scientific), according to manufacturer's protocol. To measure the infarct volume, 40-µm thick coronal sections were cut from wild-type mouse brains 7 days after PT and then counterstained with hematoxylin/eosin, dehydrated in a graded ethanol series, cleared in xylene, and cover-slipped with Permount (Fisher Scientific).

Briefly, sections were permeabilized with PBS-triton 0.1% (14-µm-thick sections) or PBS-triton 0.3% (30-µm-thick sections) for 30 min, blocked in 5% BSA for 1 h and finally incubated overnight with primary antibody in PBS with 1% BSA. Primary antibodies used were rabbit anti-Iba1 (1:500, Wako, 019-19741), goat anti-CD31 (1:500, R&D Systems, AF3628-SP), guinea pig anti-Doublecortin (1:1000, Millipore, AB2253), rat anti-GFAP (1:2000, Invitrogen, 13-0300), goat anti-GFP (1:2000, Abcam, ab5450), rabbit anti-GFP (1:2000, Abcam, ab290), sheep anti-Fibrinogen (1:500, US Biological, F4203-02F), goat anti-Nestin (1:200, Santa Cruz, sc-21249), goat anti-Nestin (1:500, Antibodies-Online, ABIN188165), rabbit anti-TMEM119 (1:500, Abcam, ab209064), rat anti-CD68 (1.200, Biorad, MCA1957), rabbit anti-C1qb (1:200, Abcam, ab182451), rabbit anti-P2ry12 (1:500, Sigma Life Science, HPA014518), rabbit anti-Sox2 (1.2000, Abcam, ab97959), rabbit anti-Mash-1/Achaete-scute (1:500, Abcam, ab74065), rabbit anti-RFP (1:2000, Rockland, 600-401-379), mouse anti-iNOS (1:200, BD Bioscience, 610329), rabbit anti-Gas6 (1:25, R&D Systems, AF986), rat anti-Ki67 (1:200, Invitrogen, 14-5698-82), goat anti-ApoE (1:50, Invitrogen, PA1-26902), rabbit anti-ApoE (1:50, Abcam, ab183597), rat anti-CD45 (1:100, BD Pharmingen, 553079), mouse anti-EGFR (1:100, Merck Millipore, 05-1047), rabbit anti-Lrp8 (1:50, ThermoFisher, bs-6651R), rabbit anti-NeuN (1:500, Abcam, ab177487), mouse anti-Olig2 (1:200, Merck Millipore, MABN50), rabbit anti-S100β (1:2000, Abcam, ab868). Secondary antibodies used included donkey antibodies to rabbit, rat, guinea pig, mouse, sheep, and goat conjugated with Alexa Fluor 488, 594, or 405 (1:200, Jackson ImmunoResearch Laboratories). Sections were cover-slipped with DAPI (Southern Biotechnology).

## TUNEL and ApopTag assays

TUNEL assays were carried out using the In Situ Cell Death Detection Kit (Roche), according to the manufacturer's instructions. Briefly, specimens were permeabilized with PBS containing 0.3% Triton-X 100 for 30 min and subsequently incubated with the TUNEL reaction solution mixture in a humidified 37 °C chamber for 1 h. For co-labeling, specimens were washed three times with PBS containing 0.1% Triton X 100, followed by blocking and primary and secondary antibody treatments as described above. ApopTag assays were carried out using the ApopTag Red In Situ Apoptosis Detection Kit (Merck) according to the manufacturer's instructions with a few modifications. Briefly, specimens were permeabilized in PBS containing 0.3% Triton-X 100 for 15 min and subsequently incubated with ethanol:acetic acid (in a proportion of 2:1) at − 20 °C for 5 min, followed by blocking and then primary and secondary antibody treatments, as above. Thereafter, specimens were treated with equilibration buffer for 10 min at RT, followed by treatment with a working strength TdT enzyme mixture for 1 h at 37 °C in a humid chamber. The enzymatic reaction was stopped with the working-strength stop/wash buffer supplied with the kit. Specimens were then treated with anti-digoxigenin conjugate for 30 min at 37 °C in a humid chamber, followed by washing steps and mounting with DAPI-Fluoromount.

## Microscopy

For mouse tissue, z-stack images (0.5–1 µm per stack) were acquired on a Leica TCS SP8 laser confocal microscope with 20 ×, 40 ×, or 63 × oil-immersion objectives. Representative images were acquired by combining Z-stack images into a maximum-intensity projection with the LAS AF image-analysis software (Leica). All imaged sections (3–4 sections per brain) belonged to the plane of injury. For the overview images of the lesion core and SVZ, a tile-scan image was acquired. Cell number counts and colocalization in the SVZ and Cortex: For quantification of the number of cells in the SVZ, a volumetric area of approximately 290 × 72 × 30 µm in the ipsilateral hemisphere of the lateral SVZ was imaged per section. For quantification of the number of cells in the cortex, a volumetric area of approximately 290 × 290 × 30 µm was imaged per section, followed by axis clipping and rotation of the 3-D-rendered images. Colocalization of different markers was analyzed with LAS AF analysis software by displaying the z-stacks as maximum intensity projections and using axis clipping and rotation of the 3-D-rendered images. Images were generated from z-stack projections (0.35 µm per stack) through a complete cortical lesion area. For quantification of RFP+Iba1 +, RFP+Iba1^low, and RFP+Iba1− cells in the SVZ and the cortex of *HexB-tdTomato* mice and in the SVZ of *Cx3cr1-CreER^T2:R26-tdTomato* mice, each RFP + microglial cell was categorized according to its level of Iba1 expression. For microglia–blood vessel contacts: An area of 290 × 72 × 30 µm in the ipsilateral hemisphere of the lateral SVZ was imaged per section. For quantification of the area of overlap of microglia with blood vessels, Cx3cr1-GFP + CD31 + colocalized areas were analyzed. For immunoreactivity analysis: An area of 290 × 72 × 14 µm in the ipsilateral hemisphere of the lateral SVZ was imaged. The maximal projected signals were saved as TIFFs. In ImageJ (NIH), the images were converted into black-and-white 8-bit images and thresholded. The total IR was calculated as area density, defined as the number of pixels (positively stained areas) in the imaged field. ApoE and CD68 immunoreactivity were calculated for all outlined SVZ microglia and are expressed as a percentage of the area of Iba1+ or Iba1−RFP + microglia, respectively. ApoE and Lrp8 signal analysis: To quantify ApoE and Lrp8 signal intensity in SVZ NSPCs, every Cx3cr1-GFP + and Nestin-TdTomato+ nuclei were individually selected and quantified. Calculated immunoreactivity values from individual NSPCs expressing Lrp8 and ApoE were averaged. The representative images were acquired with a Leica TCS SP8 laser confocal microscope with a 40 × oil-immersion objective. Images were generated from z-stack projections (0.35–1.0 µm per stack) through a distance of 1 µm. ApoE and Lrp8 proximity analysis:

First, ApoE + RFP + cells and Lrp8 + DCX + cells were identified with the Z-stack IMARIS function. Cells were reconstructed using the surface mode. IMARIS provided the smallest distance between reconstructed cells. To demonstrate the presence of Lrp8 in DCX + cells, the spots mode was used to quality-check the signal. Neuroblast migration towards the lesion border: An area of $775 \times 775\,\mu m$ was imaged per section, to simultaneously show part of the SVZ and the lesion core. DCX + Sox2 − cells that had migrated from the SVZ toward the lesion core were counted manually.

## Infarct volume quantification

The ischemic lesion was clearly delineated as a region of pallor within the cortex. The infarct was outlined with Image J on coronal sections spaced 240-μm apart, as previously described[64]. Infarct volumes were derived by calculating the average infarct area between slices and multiplying by the distance between the start and end slices.

## Three-dimensional microglia reconstruction

Free-floating 30-μm cryosections from brains of *Cx3cr1-EGFP* mice were permeabilized with PBS-triton 0.3% for 30 min, blocked in 5% BSA for 1 h and finally incubated overnight with rabbit anti-GFP (1:200, Abcam) in PBS with 1% BSA, followed by Alexa Fluor 488/594-conjugated secondary antibody (1:200, Jackson ImmunoResearch Laboratories). Sections were cover-slipped with Fluoromount containing DAPI (Southern Biotechnology). Images were acquired with a Leica TCS SP8 laser confocal microscope with $40 \times$ or $60 \times$ oil-immersion objectives. Z-stacks were collected with a 0.5–1.0 μm step size at $1024 \times 1024$ pixel resolution, and then images were analyzed in IMARIS software (Bitplane) and further analyzed as reported, with modifications[65]. Three-dimensional reconstruction was performed using IMARIS Filament Tracer with no loops allowed and spot-detection mode to determine the start and end points of microglial process extensions. Although the analysis was performed automatically by the IMARIS software, we separately verified that each microglial process originated from one defined cell and manually removed false connections. In addition, IMARIS Surface Rendering was implemented to render the microglial soma separately.

## Statistical analysis

Data are shown as the means ± SEM. Differences between groups were examined with one-way ANOVAs for multiple comparisons, followed by Bonferroni corrections for comparison of means; differences between two experimental groups were assessed by non-parametric unpaired Mann–Whitney tests or unpaired two-tailed Student's *t* tests. Analyses were conducted in GraphPad Prism (GraphPad Software, Version 6.0, La Jolla, USA). Differences were considered significant when the *P*-value was < 0.05.

## Reporting summary

Further information on research design is available in the Nature Portfolio Reporting Summary linked to this article.

# Data availability

The scRNA-Seq data generated in this study have been deposited in the NCBI Gene Expression Omnibus database under accession code GSE275939. Source data are provided in this paper.

# Code availability

The code used to analyze the data in this study has been deposited under Github [https://github.com/SchachtrupLab/Neuromac_B1.git].

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

## Acknowledgements

We thank Meike Ast-Dumbach for technical assistance, A. Schober for graphics, Gary Howard for editorial assistance, and the University of Freiburg Live Imaging Center (LIC) for microscopy support. This study was supported by the Fill in the Gap fellowship (Medical Faculty Freiburg) to S.N. and J.-D.L., the Hannelore Kohl fellowship to J.C.M.S., the Chinese Studies Program fellowship to K.W., the Müller Fahnenberg Foundation (Medical Faculty Freiburg), the European Stroke Research Foundation (ESRF), the Else Kröner-Fresenius-Stiftung 2019_A53, the DFG SFB/TRR167 subproject grant, and the DFG grants SCHA 1442/8-1, SCHA 1442/8-3, and SCHA 1442/9-1 to C.S.

## Author contributions

S.N. performed the majority of the experiments. J.S.C. and Y.-H.C. contributed to the scRNA-Seq analysis, P.C., S.S.D., and J.C.M.S. contributed to the surgeries, R.S., L.A., and Y.-H.C. performed the SVZ cell FACS isolation and scRNA-Seq, P.C., K.W., J.-D.L., and J.C.M.S. performed the immunohistochemical analysis of tissue sections, C.G. performed the acute slice imaging, A.V., M.P., and J.K.L. contributed to the experimental design, data analysis, and interpretation. C.S. designed the study, analyzed the data, coordinated the experimental work, and wrote the manuscript, with contributions from all authors.

## Funding

## Competing interests

The authors declare no competing interests.
