## [Peer Review File · Nature Communications]

Interaction between subventricular zone microglia and neural stem cells impacts the neurogenic response in a mouse model of cortical ischemic strokeREVIEWER COMMENTS

Reviewer #1 (Remarks to the Author):

The manuscript entitled, "Cross-communication between neural stem/precursor cells and microglia regulates neurogenic responses in the subventricular zone after stroke" by Nath et al seeks to examine changes within the largest neurogenic zone of the brain following photothrombotic ischemia. The manuscript uses single cell RNA sequencing to investigate changes within NSPCs and microglia following stroke. Overall, the manuscript presents an important problem and set of questions with clinical relevance. The potential impact of this study is defining the temporal changes that occur following stroke which may aid in therapeutic strategies. The study is largely descriptive with important validation of sequencing findings; however, the pathophysiological significance of the associations and correlations is unclear. For example, the authors provide a compelling study of environmental intercellular interactions and changes to cellular states, however, whether these changes are deleterious and maladaptive or beneficial are unclear. Most certainly, the targeting of ApoE-Lrp-8 by genetic or pharmacological means could significantly enhance the manuscript. With that said, all data are presented logically with rigorous analyses and are an important contribution for the scientific community.

I have a few questions/comments that may help.

Comments/Questions.

The authors utilize transgenic mice for which either tamoxifen inducible CRE downstream of the nestin promoter is expressed in neural stem cells or CRE downstream of the Cx3cr1 promoter which is present within immune cells. These mice are crossed to inducible fluorescent protein transgenic mice. Labeled cells were isolated by FACS and subject to mCEL-Seq2 using ~1500 cells. As expected, the authors identify neural lineages and microglia lineages. The neural lineage contains canonical markers that allow for the division of cells into clusters representing quiescent and activated NSPCs, TACs, and neuroblasts. The authors found increased TACs expressing proliferative markers and demonstrated an increase in EdU labeling of Sox2/GFAP B cells and Sox2 only B and TACs. Provocatively, they demonstrate that Bcl2/Gas6 which are associated with survival/apoptosis decisions are elevated and that ApopTag induced apoptosis.

Comment 1. Did the authors detect changes in the proportion of quiescent to activated NSCs after stroke?

The authors additionally indicate microglia can be clustered into 5 categories. These categories segregate into five groups and the proportion of these changes after stroke.

Comment 2. Do any of these groups represent M1 or M2 types? Is this what is meant by "The microglia clusters....related to microglia activation" (Page 6 line 107)?

Comment 3. Did the total number of microglia increase?

Comment 4. Did the authors detect changes in proliferative or apoptotic indicators?

Comment 5. Did the authors detect the presence of neural RNA fragments inside of microglia indicating phagocytosis?

Comment 6. Did any of these cells express Hoxb8, CCR2, or other potential canonical markers of additional microglia progenitors?

The authors subsequently demonstrate using HexB-tdTomato mice, that many Hexb cells do not express Iba1. Further, they demonstrate that the percentage of Cx3cr1-CRE-ERT2 derived and Iba1

double labeled cells is quite significant and that this increases after PT.

Comment 7. Are these microglia progenitors?

Comment 8. Were there differences in the number of microglia outside of the V-SVZ different-for example in the ventricle or choroid plexus? Interestingly, some of the cells appear to be on the luminal side of the V-SVZ.

The authors conclude by examining the ligand receptor pairs found within the NSPCs and microglia and identify that ApoE (microglia) and Lrp8 (NSPCs)

Reviewer #2 (Remarks to the Author):

The research paper by Nath and colleagues reveals novel aspects of cross-communication between microglia and neural stem/precursor cells (NSPCs) after photothrombotic stroke. The authors provide extensive single cell sequencing data sets suggesting that microglia regulate the neurogenic response and also identify NSPC–microglia ligand–receptor pairs that could be involved in this process. In particular, they suggest the possible role of Lrp8–ApoE interactions in NSPC–microglia cross-communication.

Overall, this is an interesting paper with potentially important data and some novel observations concerning NSPC–microglia interactions after stroke. The single cell sequencing workflow is well-executed and the analysis concerning interactions between NSPC–microglia ligand–receptor pairs is also interesting. However, there are some technical issues the authors need to clarify to make this paper sound. In addition, I greatly miss studies linking the observed microglia-NSPC interactions with the impact of these processes on functional changes in the neurogenic niche, numbers of newly born neurons, integration of them into neuronal networks and overall outcome / functional recovery after stroke.

Major points:

- Photothrombotic ischemia and tissue isolation: No histological staining has been provided to delineate the extent of the infarct and its location compared to the site of tissue isolation for single cell RNA sequencing and other measurements. Please give details on the volume of cerebral ischemia, the affected cortical layers and assess cell viability / injury in the SVZ after stroke. The boundary zone of the infarct in relationship with the SVZ should be histologically verified. In the absence of this data interpretation of the studies executed remains difficult. „Cortical stroke alters the environment in the SVZ stem cell niche (Extended Data Fig. 1a).” Please give detailed assessment of the changes suggested. It would be important to clarify whether the SVZ stem cell niche becomes ischemic in this model, would BBB injury take place, and to what extent microglial activity changes in this area compared to the core/penumbra zones of the infarct itself.

- Do the authors observe any changes in cerebral blood flow in the SVZ after photothrombotic ischemia?

- „We isolated SVZ-derived YFP+ NSPCs and CD11b+CD45^{low} microglia by flow cytometry (Extended Data Fig. 1c)” – The authors have used Nestin-CreERT2:R26-yfp mice for NSPC isolation, which may yield a broad YFP expression, and would not be specific for NSPCs only. Please give details of the cell

types expressing YFP in these mice, including astrocytes, neurons, DCX-positive cells and oligodendrocytes. Although downstream subclustering analysis appears to have identified SVZ NSPCs and excluded other, functionally distinct, YFP-positive cells, the SVZ NSPC population seems rather heterogeneous. It would also be important to know, to what extent SVZ neuroblasts were represented by YFP-positive cells. What was the proportion of DCX-positive cells that were YFP-negative?

- In general, I miss low resolution histological images and spatial information to link microglial activity changes with NSPC proliferation, migration and phagocytosis. Please provide images of the cerebral cortex from the area of infarct to the subventricular zone and display microglial Td-Tomato / Iba1 and NSPCs on the same images. It would be important to link these exciting transcriptomic data sets with spatial information at least at the level of microglia-NSPC interactions better.

- It supports the authors hypothesis concerning the origin of recruited microglia that Ccr2-RFP cells after cortical infarction were not recruited. However, CD45 immunostaining would also be suggested to assess the presence of leukocytes in the SVZ. Note that only a subset monocytes would express Cc2r among those recruited into the brain. In particular, the phenotype of Iba1-negative microglia and their origin could be interesting to further elaborate upon. Would these microglia have higher CD45 levels, and/or would these show proliferative activity?

- Microglia depletion studies: The phenotypic characterisation of microglia to eliminate dying DXC-positive cells is interesting. However, I greatly miss insight into the functional consequences of these events. Do the authors have data to show whether the numbers of newly born neurons were altered by the absence of microglia, was neuroblast migration, proliferation, etc, affected? Did microglia depletion have any impact on the wiring of cortical circuits, did it affect functional recovery, cognition or motor function? Without providing such data the observed changes remain largely descriptive.

- Methods: „PT was performed 5 days after the start of the PLX5622 diet and mice were sacrificed 7 days after PT”. Did PLX5622 treatment alter the SVZ in any means prior to stroke or influence infarct size, BBB injury, or astrocyte responses after stroke?

- The authors identify the ligand–receptor pair Lrp8–ApoE in NSPC–microglia cross-communication, which I find important in this context. However, nothing is known about the impact of these interactions on neurogenesis or functional recovery after stroke in this model. Have the authors considered using at least Lrp8 KO mice or targeting the SVZ compartment with pharmacological tools (e.g. neutralizing antibodies, blocking peptides against Lrp8) to study these processes in vivo?

Minor point:

- I would update graphics and text to name C cells and B cells „Type C cells” and „Type B cells” to ensure that readers outside the SVZ field understand their relationships and origin.

Reviewer #3 (Remarks to the Author):

In this study, the authors aim to address a neurogenic response after stroke that is claimed to be regulated by cross communication between specific types of microglia (MG) and neural stem/progenitor (NSPC) subclusters. Despite the relevance of this topic, in my opinion, the work shown here suffers from a general lack of focus and novelty.

The title and the abstract are misleading as the biological question, the interaction between MG and NSPC and its effects on neurogenic responses, is not functionally addressed throughout the whole study. The authors claim to find an interaction between MG and NSPC, but fail to show that this connection regulates the neurogenic response. Essential experiments are missing to draw this conclusion and the data presented is not sufficient to make such a statement. In addition, the quality of the data is worrisome, since in suppl. figure 2 a very high % of mitochondrial genes are shown, suggesting that sequenced cells might be apoptotic/cell death.

Along with this, their proof of MG and NSPC interaction is also rather weak and does not go beyond established knowledge in the field. That MG endocytose apoptotic cells are not astonishing, but rather depict what MG are known for. Thus, they are not novel and don't add supporting evidence for their hypothesis regarding the interaction or regulation of the neurogenic response. In sum, the current study consists mainly of repetitions and reproductions of previous studies, for instance the heterogeneity of microglia in the brain and does not add any novel insights or findings into the field of MG and NSPC regulation.

In summary, this study provides neither a new data set nor a new tool and does not present compelling evidence to support its conclusions. It also lacks a functional analysis that would have shed light on cross-communication in the subventricular zone (SVZ) during injury.

Other specific concerns are as follows:

General remarks on the manuscript:

- Line 65: provide an explanation of what YFP labels for and explain the mouse models.
- Line 138: this conclusion is not new
- Line 213-216: to support the statement, the authors should offer a quantification of the cell's proximities, as showing a single image is not very conclusive.

Figures and Analysis:

The text and figure captions mention that scRNAseq data has been visualized using Diffusion maps, but the figures show UMAP coordinate systems while the methods don't mention diffusion maps. Please clarify what method was used where.

Figure 1F-G: the quantification should be done as the ratio of apoptotic cells DCX-Apoptag positive vs all NBs or Mash1 cells. As it is, the single fact that there is a higher NB abundance at 7 dpi could show the same result, without altering NB probability to undergo apoptosis.

Figure 2A: the reviewer cannot see the continuum between cluster 0 and 1 and it is not clear why two clusters (2,3) are not present.

Figure 3: the MG-SVZ NSCs interaction based on immunohistochemistry is too weak to draw the cross-connection conclusion.

The reported size of scale bars in microscopy images seems unrealistic as they are in picometers and the labels of the scales must be separated clearer.

Figure 4: The reporting summary states that test statistics for null hypothesis testing have been reported, but no test statistics are provided.

Further on extended Figures:

Figure 1A: remove results citation from the introduction.

Figure 1b: misspelling of dissociation.

Figure 2: the % of mitochondrial genes is very very high, suggesting that sequenced cells might be apoptotic/cell death.

Figure 3: given the small size and low resolution of the images, it is hard for the reader to observe double positive cells. In addition, the signal seems overexposed.

Figure 4: the authors did not consider CreERT2 inefficient recombination. And the HexB and Cx3cr1 strategy does not add on microglia heterogeneity when already scRNA-seq data is providing this information.

Point-by-Point Response

Our manuscript entitled “**Interaction between subventricular zone microglia and neural stem cells impacts the neurogenic response after stroke**” has been reviewed by three experts in the field and, overall, all three emphasized the importance of the study. Reviewer 1 stated that the “*manuscript presents an important problem and set of questions with clinical relevance*” and “*all data are presented logically with rigorous analysis and are an important contribution for the scientific community*”. Reviewer 2 stated that the manuscript “*reveals novel aspects of cross-communication between microglia and neural stem/precursor cells (NSPCs) after photothrombotic ischemia*”. Furthermore, Reviewer 2 stated that “*this is an interesting paper with potentially important data and some novel observations concerning NSPC-microglia interactions after stroke. The single cell sequencing workout is well executed and the analysis concerning interactions between NSPC-microglia ligand-receptor pairs is also interesting*”. Reviewer 3 wrote that “*despite the relevance of this topic, in my opinion, the work shown here suffers from a general lack of focus and novelty*”.

We thank the reviewers for their insightful and constructive comments and suggestions for further experiments. All of the reviewers’ points were very well taken. All three reviewers asked for further experiments to back up the appropriateness of the technical approaches, to confirm the robustness of the data from which we drew our conclusions, and to link the observed microglia–NSPC interactions with the impact of these processes on functional changes in the neurogenic niche. We completely agree with the reviewers and editor that further experiments were necessary to confirm that microglia and NSPC interactions in the SVZ regulate neurogenic responses after stroke.

Prompted by the constructive and insightful comments of Reviewers 1, 2, and 3, we performed additional experiments to further characterize the SVZ microglia–NSPC interaction after photothrombotic ischemia (PT).

In particular, we experimentally addressed the two major points made by the reviewers, namely:

- 1) Back up the appropriateness of the technical approaches and confirm the robustness of the data from which we drew our conclusions.**
- 2) Address the pathophysiological significance of SVZ microglia–neural stem cell interactions after PT.**

Here, we summarize the changes made in our revised manuscript in response to these two important points:

- 1) We added further experimental evidence to validate the suitability of the technical approaches and to confirm the robustness of the data from which we drew our conclusions:**
 - To better delineate the extent of the cortical lesion injury, to indicate the area of analysis in the SVZ stem cell niche, and to confirm early SVZ microglial activation after PT, we performed new experiments and provide new low-magnification histological images and quantification (**Fig. 1a, Extended Data Fig. 1a** in the revised manuscript). The cortical injury induced by PT results in a defined, reproducible cortical lesion that does not damage the SVZ stem cell niche. However, the distant cortical injury results in vasculature changes in the SVZ stem cell niche, as we previously reported [1] (fibrinogen leakage into the distant SVZ stem cell niche environment) and early microglial activation (**Fig. 1b and Extended Data Fig. 1b** in the revised manuscript), which led us to investigate in detail the SVZ microglia–NSPC interaction and the impact of this interaction on the neurogenic response after PT (**Fig. 1c** in the revised manuscript).
 - To validate the suitability of the *Nestin-Cre* mouse line for NSPC analysis, we further demonstrated the specificity of the *nestin* promoter in the CNS, showing exclusive labeling of SVZ NSPCs (**Extended Data Fig. 1c-e** in the revised manuscript). To validate the isolation of SVZ microglia for the downstream scRNA-Seq analysis, we performed new experiments to further evaluate the infiltration of CD45⁺ leukocytes in the SVZ after PT (**Extended Data Fig. 1h** in the revised manuscript). Although blood-derived CCR2⁺ monocytes and CD45⁺ leukocytes were found to barely infiltrate the SVZ stem cell niche after cortical PT (**Extended Data Fig. 1g-h** in the revised manuscript), we excluded these cell populations (CD11b⁺CD45^{high}Ly6C/Ly6G⁺) in the gating

strategy to obtain a pure SVZ microglial population (**Extended Data Fig. 1f** in the revised manuscript).

- To further characterize the SVZ microglial and NSPC behavior after PT, and to validate our conclusions from the downstream subclustering analysis of the scRNA-Seq data, we performed new experiments and analyses. Immunolabeling demonstrated an increased proportion of activated type B cells (**Extended Data Fig. 3c** in the revised manuscript) and activation of microglia at day 1 after PT (**Fig. 1b, Extended Data Fig. 1b** in the revised manuscript), and we also observed microglial phagocytosis of neuroblasts at day 7 after PT (**Extended Data Fig. 5c** in the revised manuscript).
- To further characterize the SVZ microglia–NSPC ligand–receptor pairs after PT, we analyzed the proximity of Lrp8+ NSPCs to ApoE+ microglia using 3-D IMARIS-processed images. We observed close contact between Lrp8+DCX+ neuroblasts and RFP+ microglia and between ApoE+RFP+ microglia and DCX+ neuroblasts in the SVZ of mice 7 days after PT (**Fig. 3e-f** in the revised manuscript). Our results show a drastically changed SVZ microenvironment after PT, with increased blood-derived fibrinogen deposition and perivascular microglia cluster formation (**Fig. 2a** in the revised manuscript). Pharmacological depletion of fibrinogen alleviated the stroke-induced SVZ environmental change and reduced SVZ ApoE+ microglia (**Fig. 3g** in the revised manuscript), potentially explaining the increased SVZ-derived neurogenic response in fibrinogen-depleted mice after PT (**Fig. 2d** in the revised manuscript). Future studies will genetically or pharmacologically target our proposed repository of SVZ microglia–NSPC ligand–receptor pairs after stroke.

2) We further experimentally addressed the pathophysiological significance of SVZ microglia–NSPC interactions after PT:

Upon cortical stroke, the SVZ stem cell niche environment is drastically changed with increased SVZ vasculature permeability and fibrinogen deposition [1] (**Fig. 2a** in the revised manuscript), coinciding with early SVZ microglial activation (**Fig. 1b, Extended Data Fig. 1b** in the revised manuscript). Fibrinogen has been demonstrated to activate microglia [2,3].

To analyze the **pathophysiological significance of SVZ microglia–NSPC interactions** after PT, we administered anrod to acutely deplete fibrinogen from the circulation and counted the number of SVZ-derived DCX+ neuroblasts in the peri-infarct lesion area in the PT mouse model of stroke. Anrod-treated animals showed a 93% reduction of fibrinogen levels in the brain parenchyma compared with vehicle-treated control animals (**data not shown**). Acute fibrinogen depletion by the pharmacological reagent anrod resulted in a 60% reduction in activated microglia in the SVZ at 10 days post-injury, compared with control mice (**Extended Data Fig. 5e** in the revised manuscript). Importantly, fibrinogen depletion increased the SVZ-derived DCX+ neurogenic response in the lesion penumbra by ~2-fold (**Fig. 2d** in the revised manuscript).

Next, we analyzed how depletion of SVZ microglia affects neuroblast formation and survival. Depletion of microglia with the colony-stimulating factor 1 receptor (CSF1R) inhibitor PLX5622 reduced cell death of DCX+ neuroblasts and increased the number of newborn EdU+DCX+ neuroblasts in the SVZ 7 days after PT, increasing the overall number of DCX+ cells (**Fig. 2e** and **Extended Data Fig. 5f-h** in the revised manuscript).

Overall, our data show that microglial depletion increased the formation and survival of newborn neuroblasts in the SVZ and the pharmacological restoration of the niche environment increased the SVZ-derived neurogenic repair after PT. In the revised manuscript, we have added five new figure panels (**Fig. 2d-e, Extended Data Fig. 5e-h**) and new text to the Results (**page 8, lines 165-173, and pages 8-9, lines 188-200**) and Discussion (**pages 12-13, lines 269-305**), describing how activated microglia in the SVZ reduced the SVZ-derived neurogenic response for brain repair. Furthermore, we have demonstrated that pharmacological depletion of fibrinogen alleviated the stroke-induced environmental changes in the SVZ and reduced the number of SVZ ApoE+ microglia (**Fig. 3g** in the revised manuscript), potentially explaining the increased SVZ-derived neurogenic response in fibrinogen-depleted mice after PT (**Fig. 2d** in the revised manuscript). These new data demonstrate the pathophysiological significance of SVZ microglia–NSPC interactions on the neurogenic response after PT.

Overall, these results reveal that changes in the interaction between SVZ microglia and NSPCs have pathophysiological relevance, greatly affecting the formation and survival of newborn neuroblasts in the SVZ and their ectopic contribution to the peri-infarct lesion area. In particular, we show that

discrete stroke-associated SVZ microglial states negatively impact the neurogenic response, and we propose a repository of relevant microglia–NSPC ligand–receptor pairs.

Overall, we changed the title of the manuscript, we added 23 new Figure panels (**Fig. 1a–c, Fig. 2d, Fig. 3e–h, Extended Data Fig. 1a–e, 1h, 3c, 4d, 5c, 5e–g, 5i–j, 6c**), revised **Fig 1j, Fig. 2a–b, Fig. 2e** and **Extended Data Fig. 3h, 4b–c, e–f, 5a**, added new text to the Summary, Results, Methods, and Discussion, and added new references.

We would like to mention that the revision process of this project was severely delayed due to COVID-19-related circumstances.

Our new results bolster our original finding that microglia, as an integral part of the distinctive SVZ microenvironment, control NSPC behavior after stroke. Our results highlight that distant brain injuries induce immune responses in the SVZ niche and alter cell–cell communication, with potential implications for the identity of SVZ cells responding to stroke and other CNS diseases beyond the newborn neurons examined here.

We believe that our study will provide a basis for future studies to uncover microglia–NSPC ligand–receptor interactions in the healthy CNS and during CNS disease. Our repository of ligand–receptor pairs will help researchers elucidate the biology of microglia and NSPCs. A detailed point-by-point response to all of the reviewers’ comments follows below, and all changes in the revised manuscript are marked in blue.

Reviewer #1:

Comments for the Author:

The manuscript entitled, “Cross-communication between neural stem/precursor cells and microglia regulates neurogenic responses in the subventricular zone after stroke” by Nath et al seeks to examine changes within the largest neurogenic zone of the brain following photothrombotic ischemia. The manuscript uses single cell RNA sequencing to investigate changes within NSPCs and microglia following stroke. Overall, the manuscript presents an important problem and set of questions with clinical relevance. The potential impact of this study is defining the temporal changes that occur following stroke which may aid in therapeutic strategies. The study is largely descriptive with important validation of sequencing findings; however, the pathophysiological significance of the associations and correlations is unclear. For example, the authors provide a compelling study of environmental intercellular interactions and changes to cellular states, however, whether these changes are deleterious and maladaptive or beneficial are unclear. Most certainly, the targeting of ApoE-Lrp-8 by genetic or pharmacological means could significantly enhance the manuscript. With that said, all data are presented logically with rigorous analyses and are an important contribution for the scientific community.

We thank the reviewer for raising the important point on the significance of SVZ microglia–NSPC interactions. We further experimentally addressed the pathophysiological significance of SVZ microglia–NSPC interactions after PT, as already summarized at the start of this response under “**Major points of the Reviewers**”.

Here, we are investigating how upon a remote cortical brain injury (cortical PT), the altered SVZ niche environment (with blood-derived fibrinogen deposition) is leading to immediate activation of SVZ microglia and to the abnormal neurogenic response, with cell-cycle arrest of neural stem cells and neuroblast cell death. Overall, our data suggest that the altered niche environment leads to discrete stroke-associated SVZ microglial clusters (cluster 1, cluster 4) that negatively impact SVZ neurogenesis and that the altered cross-communication between microglial subclusters and NSPCs regulates the extent of the innate neurogenic repair response in the SVZ after stroke.

As the Lrp8 depleted mouse line (JAX #003524) has been described to have several brain developmental defects [4], the conditional knockout mouse line was not available and as intracerebroventricular injection approaches will potentially affect the SVZ microglia – NSPC interaction (Our lab has shown fibrinogen deposition in the SVZ after cortical stab wound injury, please see Pous *et al.* [1], supplementary Figure 1), we decided to pharmacologically restore the niche environment (fibrinogen depletion by anicrod) and to deplete SVZ microglia to investigate the pathophysiological significance of the SVZ microglia – NSPC interaction after PT. Briefly, our new data revealed that depletion of fibrinogen reduces SVZ microglial activation and increases the number of DCX+ neuroblasts in the lesion penumbra. Depletion of microglia

increases the number of newborn neuroblasts and reduces neuroblast cell death in the SVZ stem cell niche (**Fig. 2d-e, Fig. 3g, Extended Data Fig. 5e-h** in the revised manuscript).

To further describe the microglia-NSPC ligand-receptor interaction (as already summarized at the start of this response under “**Major points of the Reviewers**”), we performed new experiments and analyzed the proximity of Lrp8+ NSPCs to ApoE+ microglia using 3-D IMARIS-processed images. This revealed close contact between Lrp8+DCX+ neuroblasts and RFP+ microglia and between ApoE+RFP+ microglia and DCX+ neuroblasts in the SVZ of mice 7 days after PT (proximity close to 0 μm) (**Fig. 3e-f** in the revised manuscript). Furthermore, depletion of fibrinogen reduces the number of SVZ ApoE+ microglia (**Fig. 3g** in the revised manuscript), suggesting that indeed stroke-induced SVZ microglial cluster regulate the extent of the innate neurogenic repair response in the SVZ by altered microglia-NSPC ligand–receptor interactions after stroke.

An emerging concept is the reciprocal interaction between inflammatory cells and NSPCs in CNS disease [5]. As microglia–NSPC interactions in the healthy CNS and during disease are still poorly defined, our repository of ligand–receptor pairs will help elucidate the biology of microglia – NSPC interactions in future studies. We added sentences to the Discussion section, **pages 15, lines 335-343** in the revised manuscript to emphasize that our repository of ligand–receptor pairs will help to elucidate the biology of microglia and NSPCs interaction, which will be addressed in future studies. Furthermore, we changed the title of the manuscript to better focus the manuscript on how the altered SVZ niche environment leads to discrete stroke-associated SVZ microglial clusters that negatively impact SVZ neurogenesis.

I have a few questions/comments that may help.

Major Comments

The authors utilize transgenic mice for which either tamoxifen inducible CRE downstream of the nestin promoter is expressed in neural stem cells or CRE downstream of the Cx3cr1 promoter which is present within immune cells. These mice are crossed to inducible fluorescent protein transgenic mice. Labeled cells were isolated by FACS and subject to mCEL-Seq2 using ~1500 cells. As expected, the authors identify neural lineages and microglia lineages. The neural lineage contains canonical markers that allow for the division of cells into clusters representing quiescent and activated NSPCs, TACs, and neuroblasts. The authors found increased TACs expressing proliferative markers and demonstrated an increase in EdU labeling of Sox2/GFAP B cells and Sox2 only B and TACs. Provocatively, they demonstrate that Bcl2/Gas6 which are associated with survival/apoptosis decisions are elevated and that Apoptag induced apoptosis.

Q1. Did the authors detect changes in the proportion of quiescent to activated NSCs after stroke?

We thank the reviewer for this insightful question. To address the reviewer’s concern and to boost our conclusions from the downstream subclustering analysis of the scRNA-Seq data, which revealed that the proportion of activated type B cells was approximately 3.5 times higher in PT D1 mice than in uninjured mice (**Extended Data Fig. 3b** in the revised manuscript), we performed new experiments using epidermal growth factor receptor (EGFR) labeling in combination with GFAP and Sox2 labeling, to specifically label activated type B cells [6,7]. Our new data show a ~10-fold increase in the number of EGFR+GFAP+Sox2+ cells at day 1 after PT compared to uninjured mice (**Extended Data Fig. 3c** in the revised manuscript), confirming our scRNA-Seq data and indicating that cortical stroke leads to a robust increase in activated NSPCs. We have added text to the Results section (**page 5, lines 104-109**), added a new figure panel (**Extended Data Fig. 3c** in the revised manuscript), and updated the corresponding figure legend in the revised manuscript.

Q2. The authors additionally indicate microglia can be clustered into 5 categories. These categories segregate into five groups and the proportion of these changes after stroke. Do any of these groups represent M1 or M2 types? Is this what is meant by “The microglia clusters....related to microglia activation” (Page 6 line 107)?

We thank the reviewer for this comment. We would like to emphasize that we conducted the post-sequencing analysis and cluster identification of the SVZ microglia after cortical stroke in this manuscript (five subclusters; clusters 0 to 4) using the most recent annotated SVZ microglial markers from scRNA-Seq results [8-14]. Our brief description of the individual subclusters 0 to 4 is based on comparison of the top differentially expressed gene (DEG) expression as compared to recently described scRNA-Seq microglial marker expression [8-14]. Current research examining the transcriptome, proteome, and epigenome of

microglia under various disease/injury contexts with cutting-edge technologies has revealed clear dynamism and heterogeneity in microglia, whereas the M1/M2 polarization scheme has failed to yield meaningful accounts of microglial states and functions [15,16]. Therefore, we believe that comparing our scRNA-Seq data with M1- or M2-type gene expression does not reflect the current state-of-the-art for microglial states.

However, as suggested by the reviewer, we explicitly queried microglial expression of genes associated with M1/M2 macrophage polarization. We referred to Table 2 of the review article by Murray et al. [17] for M1/M2 macrophage gene sets. M1 macrophage-associated genes were primarily expressed by clusters 1, 3, and 4 ('activated', 'cell-cell adhesion and migration', and 'disease-associated' microglia; **Point-by-Point Response Fig. 1**). We also checked for changes in M1 macrophage-associated genes as a function of injury time point, and observed that M1 macrophage-associated genes are detected primarily at day 1 after PT (**Point-by-Point Response Fig. 1**). When we queried M2 macrophage-associated genes, few were detected (**data not shown**). The highest expressing M2 gene was *Clec7a*, by cluster 2. Similarly, we observed no strong changes in M2 macrophage genes as a function of injury time point (**data not shown**). Overall, we conclude that, while microglia show cluster-specific and injury-dependent expression of some M1 macrophage genes, expression was not exclusive to a single subtype but was instead heterogeneous. We have added references to our brief description of the individual subclusters 0 to 4 in the revised manuscript (Results section, **page 6, lines 125-140**) to better emphasize what this description is based on.

Figure 1: M1 gene association with SVZ microglial subcluster. Dot plot showing the M1 macrophage-associated genes in the SVZ microglial subclusters (left). Dot plot showing the M1 macrophage-associated genes in SVZ microglia of uninjured mice or of mice 1 or 7 days after PT (right).

Q3. Did the total number of microglia increase? Did the authors detect changes in proliferative or apoptotic indicators?

We thank the reviewer for this important comment. We carried out additional experiments and carefully analyzed microglia cell numbers as well as microglial proliferation and apoptosis in the SVZ at different time points after PT (**Extended Data Fig. 5i-j** in the revised manuscript). Our new data reveal a tendency towards increased RFP+ microglial numbers in the SVZ at day 3 after PT in *Cx3cr1-CreER^{T2}:Rosa26-tdTomato* mice. However, we did not detect any microglial proliferation (immunolabeling for 5'-ethynyl-2'-deoxyuridine (EdU) of RFP+ SVZ microglia) (**Extended Data Fig. 5i** in the revised manuscript) or microglial apoptosis (immunolabeling for ApopTag of RFP+ SVZ microglia) (**Extended Data Fig. 5j** in the revised manuscript) in the SVZ at the different timepoints after PT. We have added this information to **page 8, lines 184-188** of the revised manuscript, suggesting that the slight increase in the number of microglia in the SVZ after PT might be because of a migratory contribution by nearby microglia.

Q4. Did the authors detect the presence of neural RNA fragments inside of microglia indicating phagocytosis?

We thank the reviewer for this insightful comment. Our data showed that microglia in the SVZ phagocytose DCX+ neuroblasts in the SVZ (**Fig. 1j** and **Extended Data Fig. 5d** in the revised manuscript). Violin plots for the C cell and neuroblast genes *Ascl1* and *Dcx* demonstrate that neural transcripts were indeed detected in microglia (**Extended Data Fig. 5c** in the revised manuscript). Furthermore, these transcripts are nearly undetectable in the unknown/ependymal cell population. As expected, these transcripts were the highest in NSPCs (**Extended Data Fig. 5c** in the revised manuscript). We have added the new results to **page 8, line 184** (Results section) in the revised manuscript. Overall, these data support our finding that microglia phagocytose NSPCs in the SVZ after stroke.

Q5. Did any of these cells express *Hoxb8*, *CCR2*, or other potential canonical markers of additional microglia progenitors? The authors subsequently demonstrate using *HexB-tdTomato* mice, that many

Hexb cells do not express Iba1. Further, they demonstrate that the percentage of Cx3cr1-CRE-ERT2 derived and Iba1 double labeled cells is quite significant and that this increases after PT. Are these microglia progenitors?

We thank the reviewer for her/his insightful comment. We carefully re-examined the SVZ microglial phenotype and fine-tuned the Iba1 (ionized calcium-binding adaptor molecule 1) expression threshold to strictly distinguish SVZ microglia with a low level of Iba1 expression from microglia with no detectable Iba1 expression (Methods section **page 30, lines 698-701** in the revised manuscript). Our revised data confirm that immunolabeling for Iba1 in uninjured *HexB-tdTomato* transgenic mice resulted in ~100% co-labeling of cortical microglia (**Extended Data Fig. 4b** in the revised manuscript). By contrast, ~40% of HexB+ cells (red) in the SVZ either had a low level of Iba1 expression or were negative for Iba1 (green) (**Extended Data Fig. 4b** in the revised manuscript). In line with this, immunolabeling for Iba1 in *Cx3cr1-CreER^{T2}:R26-tdTomato* mice at day 1 and day 7 after PT and in uninjured mice revealed that ~15%–20% of RFP+ SVZ microglia either had low expression levels or were negative for Iba1 (green) (**Extended Data Fig. 4c** in the revised manuscript).

We queried the expression of microglial progenitor-associated genes and found that *Hoxb8* and *Ccr2* were not found in SVZ microglia (**Point-by-Point Response Fig. 2A**). Furthermore, immunolabeling for IRF8, a marker for microglial progenitor cells [18], was not detected in SVZ RFP+ microglia at days 1, 3, or 7 after PT. Mouse spleen tissue was used as a positive control for IRF8 antibody function [19] (**Point-by-Point Response Fig. 2B**). A distinct Iba1- microglia population with expression of CD68 has been reported in deep subcortical brain lesions with yet unknown functions [20]. We performed new experiments and found that ~40% of SVZ-populating Iba1-RFP+ microglia expressed CD68 (**Extended Data Fig. 4d** in the revised manuscript). Overall, our data suggest that the dynamic environmental signals in the SVZ neurogenic niche may foster a higher diversity of microglial phenotypes and functionality.

In summary, no SVZ microglia expressed progenitor cell markers. As a consequence, we carefully checked the literature and changed the description of cluster 3 (originally described as expressing genes related to developmental processes) to indicate enrichment in genes related to cell–cell adhesion and migration (Results section **page 6, lines 134-136** in the revised manuscript). Together, our immunohistochemical analyses confirm a heterogeneous molecular composition of microglia in the distinct SVZ stem cell niche microenvironment and their functional changes after PT.

Figure 2: Microglial progenitors are not detected in the SVZ. a, Dot plot of selected microglial progenitor-associated genes expressed by different cell populations. **b**, Immunolabeling for IRF8 (green) in the spleen of an uninjured WT mouse. The white dashed box indicates the magnification of IRF8+ cells. Immunolabeling for IRF8 (green) and RFP (red) in the SVZ 1, 3, and 7 days after PT, compared with uninjured mice. Scale bars, 70 μ m, spleen; 17 μ m, enlargement of the spleen; 30 μ m, SVZ. Quantification of the percentage of RFP+IRF8+ cells normalized to RFP+ cells in the SVZ in uninjured mice and after PT (n = 3 mice). The graph shows the mean \pm SEM.

Q6. Were there differences in the number of microglia outside of the V-SVZ different-for example in the ventricle or choroid plexus? Interestingly, some of the cells appear to be on the luminal side of the V-SVZ.

We thank the reviewer for this interesting observation. We would like to highlight that we only analyzed SVZ stem cell niche microglia in this study. We have better outlined the area of analysis in **Fig. 1a-c** in the revised manuscript to emphasize that we analyzed microglia and NSPCs in only the SVZ stem cell niche at the different time points after PT and in uninjured mice. So far, all our experiments after cortical stroke indicate a tendency towards increased numbers of microglia in the SVZ at day 3 after PT without any SVZ microglial proliferation, apoptosis, or infiltration of monocytes into the SVZ (**Extended Data Fig. 1g-h, 5i-j** in the revised manuscript), suggesting that nearby microglia might increase the SVZ microglial density after PT.

We performed new experiments and quantified RFP+ microglia/macrophages in the choroid plexus, revealing a 3- to 4-fold increase in microglia/macrophages at 3 and 7 days after PT, respectively (**Point-by-Point Response Fig. 3**). We carefully reanalyzed our images and did not find any evidence for microglia on the luminal side of the SVZ (**data not shown**). Although we cannot rule out a contribution of microglia from the choroid plexus to the SVZ stem cell niche, we suggest that the slight increase in SVZ RFP+ microglia after PT might result from a migratory contribution of nearby microglia and added text to the Result section, **page 8, lines 184-188** in the revised manuscript.

Figure 3. The number of microglia/macrophages in the choroid plexus increases after PT. Immunolabeling for RFP (red) in the choroid plexus 1, 3, and 7 days after PT, compared with uninjured mice. Scale bar, 60 μ m. Quantification of RFP+ cells in the choroid plexus in uninjured mice and at different time points after PT (n = 3 mice). The plot shows the mean \pm SEM. ns, not significant. Differences between groups were examined with one-way ANOVAs for multiple comparisons, followed by Bonferroni corrections for comparisons of means.

Reviewer #2:

Comments for the Author:

The research paper by Nath and colleagues reveals novel aspects of cross-communication between microglia and neural stem/precursor cells (NSPCs) after photothrombotic stroke. The authors provide extensive single cell sequencing data sets suggesting that microglia regulate the neurogenic response and also identify NSPC–microglia ligand–receptor pairs that could be involved in this process. In particular, they suggest the possible role of Lrp8–ApoE interactions in NSPC–microglia cross-communication.

Overall, this is an interesting paper with potentially important data and some novel observations concerning NSPC–microglia interactions after stroke. The single cell sequencing workout is well-executed and the analysis concerning interactions between NSPC–microglia ligand–receptor pairs is also interesting. However, there are some technical issues the authors need to clarify to make this paper sound. In addition, I greatly miss studies linking the observed microglia–NSPC interactions with the impact of these processes on functional changes in the neurogenic niche, numbers of newly born neurons, integration of them into neuronal networks and overall outcome / functional recovery after stroke.

Q1. Photothrombotic ischemia and tissue isolation: *No histological staining has been provided to delineate the extent of the infarct and its location compared to the site of tissue isolation for single cell RNA sequencing and other measurements. Please give details on the volume of cerebral ischemia, the affected cortical layers and assess cell viability / injury in the SVZ after stroke. The boundary zone of the infarct in relationship with the SVZ should be histologically verified. In the absence of this data interpretation of the studies executed remains difficult. „Cortical stroke alters the environment in the SVZ stem cell niche (Extended Data Fig. 1a).” Please give detailed assessment of the changes suggested. It would be important to clarify whether the SVZ stem cell niche becomes ischemic in this model, would BBB injury take place, and to what extent microglial activity changes in this area compared to the core/penumbra zones of the infarct itself. Do the authors observe any changes in cerebral blood flow in the SVZ after photothrombotic ischemia?*

We thank the reviewer for raising these important points. We have performed new experiments and immunohistochemical staining to delineate the extent of the infarct, to show the intactness of the SVZ tissue,

and to outline the site of tissue isolation for single-cell RNA sequencing and other analyses and measurements (**Fig. 1a-c** in the revised manuscript). The new experiments to delineate the extent of the infarct volume revealed high consistency of the cortical lesion after PT (**Extended Data Fig. 1a** in the revised manuscript).

Our lab previously established that distant cortical injuries result in increased permeability of the SVZ stem cell niche vasculature and subsequent blood-derived fibrinogen leakage into the SVZ stem cell niche environment (fibrinogen is a 340 kDa blood-derived coagulation factor protein) [1]. We performed new experiments and confirmed that the SVZ niche microenvironment changes after PT: blood-derived fibrinogen is deposited in the SVZ microenvironment, leading to microglial clustering around the SVZ stem cell niche vasculature and immediate SVZ microglial activation (**Fig. 1b, Fig. 2a, Extended Data Fig. 1b** in the revised manuscript). However, immunolabeling for HIF1 α (green, a main molecule that responds to hypoxia) after PT demonstrated the absence of HIF1 α ⁺ cells in the SVZ, although many HIF1 α ⁺ cells were present in the lesion penumbra (**Point-by-Point Response Fig. 4**). We next investigated the SVZ microglia–NSPC interaction after stroke through scRNA-Seq and further analysis to better understand whether the early SVZ niche microenvironmental changes and the SVZ microglial activation might be the potential cause of the limited SVZ-derived neurogenic repair after stroke.

Overall, our new data highlight that cortical PT leads to a very consistent cortical lesion that does not damage the SVZ tissue architecture. The distant cortical injury results in increased permeability of the vasculature, with deposition of blood-derived fibrinogen into the SVZ stem cell niche environment and microglial activation. We have added new figures (**Fig. 1a-c, Extended Data Fig. 1a-b** in the revised manuscript), and new text in the Results section (**pages 3-4, lines 68-81**) to the revised manuscript. Overall, our study suggests that stroke-induced changes in vasculature permeability and in the SVZ niche microenvironment immediately activate distinct SVZ niche microglial clusters, altering the intimate interactions between SVZ immune cells and NSPCs. The underlying mechanisms by which a distant cortical injury affects the permeability of the vasculature are not yet understood.

Figure 4. No increase in HIF1 α ⁺ cells in the SVZ after PT. Immunolabeling for HIF1 α (green) 1 and 3 days after PT, compared with uninjured mice. The white dashed boxes in the overview indicate the areas of magnification for the SVZ and the penumbra. Scale bars, 350 μ m in the overview, 45 μ m in the magnifications. Quantification of the number of HIF1 α ⁺ cells in the penumbra and in the SVZ 1 and 3 days after PT, compared with uninjured mice (n = 3 mice). The plot shows the mean \pm SEM. **P<0.01, ***P<0.001, ns, not significant. Differences between groups were examined by a one-way ANOVA for multiple comparisons, followed by Bonferroni corrections for comparisons of means.

Q2. “We isolated SVZ-derived YFP⁺ NSPCs and CD11b⁺CD45^{low} microglia by flow cytometry (Extended Data Fig. 1c)” – The authors have used Nestin-CreERT2:R26-yfp mice for NSPC isolation, which may yield a broad YFP expression, and would not be specific for NSPCs only. Please give details of the cell types expressing YFP in these mice, including astrocytes, neurons, DCX-positive cells and oligodendrocytes. Although downstream subclustering analysis appears to have identified SVZ NSPCs and excluded other, functionally distinct, YFP-positive cells, the SVZ NSPC population seems rather heterogenous. It would also be important to know, to what extent SVZ neuroblasts were represented by YFP-positive cells. What was the proportion of DCX-positive cells that were YFP-negative?

We thank the reviewer for this important comment. We would like to note that the *Nestin-Cre* mouse line used in our study is the most CNS-NSPC-specific nestin line available (line *k*) [21], with restricted CNS labeling limited to adult neurogenic niches. Furthermore, the tamoxifen treatment regime of 5 consecutive injections (resulting in maximal DNA recombination) and a waiting period of 12 days before PT (for optimal tamoxifen clearance [22]) was chosen to avoid YFP expression in nestin re-expressing non-NSPCs after stroke, resulting in the most-selective targeting of SVZ NSPCs currently feasible.

To address the reviewer's concern regarding the specificity of the *nestin* promoter in the CNS, we have performed new experiments and demonstrated that, after tamoxifen treatment, the mouse line exhibited restricted gene recombination, labeling NSPCs exclusively (**Extended Data Fig. 1d-e** in the revised manuscript). We have added new text to the Results section on **page 4, lines 81-84** in the revised manuscript, mentioning that we selected the most CNS-NSPC-specific *nestin* line available (line *k*), resulting in restricted gene recombination and specific NSPC labeling following tamoxifen treatment.

Next, we performed new experiments to address the extent to which SVZ neuroblasts were represented by YFP-positive cells and to quantify the proportion of YFP+ cells that were also DCX+. We would like to emphasize that SVZ NSPCs differentiate along the neurogenic lineage (from quiescent type B cells to newborn neurons) and that the *Nestin-Cre* mouse line we used labels a low percentage of this lineage cells [23]. Our new data revealed that around 4% of all DCX+ cells were also YFP+, indicating that part of the newly formed neuroblasts were labeled with the *Nestin-Cre* reporter mouse line we used (**Extended Data Fig. 1c** in the revised manuscript). We added new text to the Results section (**page 4, lines 81-84**) in the revised manuscript, indicating that the transgenic mouse line we used had restricted gene recombination that exclusively labeled CNS NSPCs, including DCX+ neuroblasts, allowing us to investigate how microglia and NSPCs interact in the SVZ stem cell niche after stroke.

Q3. In general, I miss low resolution histological images and spatial information to link microglial activity changes with NSPC proliferation, migration and phagocytosis. Please provide images of the cerebral cortex from the area of infarct to the subventricular zone and display microglial Td-Tomato / Iba1 and NSPCs on the same images. It would be important to link these exciting transcriptomic data sets with spatial information at least at the level of microglia-NSPC interactions better.

We thank the reviewer for raising this important point. As mentioned above, we totally agree that for correct understanding of our manuscript, it is vital to differentiate the location of the cortical infarct from the area of investigation of this study, namely the SVZ. We would like to emphasize that all analyses in this study, including the scRNA-Seq analysis and the immunohistochemical analysis, were conducted with cells isolated from the SVZ or in the SVZ stem cell niche.

To better illustrate the cortical lesion and the area of analysis (the SVZ stem cell niche) in this study, we performed new experiments and now provide low-magnification images showing the cortical lesion after cortical ischemic stroke, with labeling for microglia (RFP) and NSPCs (DCX) in the same images (**Fig. 1a** in the revised manuscript). We quantified the area of the cortical lesion (**Extended Data Fig. 1a** in the revised manuscript), demonstrating that the cortical injury induced by PT results in a defined cortical lesion area that does not damage the SVZ stem cell niche.

However, the cortical injury does result in alterations in the integrity of the SVZ stem cell niche vasculature (fibrinogen leakage into the distant SVZ stem cell niche environment), as expected (**Fig. 2a** in the revised manuscript and [1]), and microglial activation (**Fig. 1b and Extended Data Fig. 1b** in the revised manuscript). This may be the cause of the limited neurogenic response towards the cortical injury site and led us to investigate in detail the SVZ microglia-NSPC interaction after PT (**Fig. 1c** in the revised manuscript).

To summarize, we have provided new low-magnification histological images and spatial information to better delineate the extent of the cortical injury after PT and to better indicate the SVZ stem cell niche as the area of analysis. In the revised manuscript, we have added new figure panels (**Fig. 1a-c, Extended Data Fig. 1a-b** in the revised manuscript) and new text to the Introduction and Results sections to emphasize that all transcriptomic data (and further analysis of microglia-NSPC interactions) were obtained from the SVZ.

Q4. It supports the authors hypothesis concerning the origin of recruited microglia that Ccr2-RFP cells after cortical infarction were not recruited. However, CD45 immunostaining would also be suggested to assess the presence of leukocytes in the SVZ. Note that only a subset monocytes would express Cc2r among those recruited into the brain. In particular, the phenotype of Iba1-negative microglia and their origin could be interesting to further elaborate upon. Would these microglia have higher CD45 levels, and/or would these show proliferative activity?

We thank the reviewer for this thoughtful comment. We performed new experiments and immunolabeling for CD45 in combination with RFP in the SVZ on day 1 after PT and found no infiltration of leukocytes in the experimental animals compared with the uninjured control (**Extended Data Fig. 1h** in the revised manuscript). This suggests that the slight increase in SVZ RFP+ microglia after PT might result from a

migratory contribution of nearby microglia to the SVZ stem cell niche after PT and not from infiltration of blood-derived monocytes or leukocytes. We have added a new figure panel (**Extended Data Fig. 1h**) and text (**page 4, lines 87-90** and **page 8, lines 184-188**) to the revised manuscript.

We agree with the reviewer that the heterogeneity of SVZ microglia, especially the Iba1-negative population, is very interesting and invites further examination. With our current scRNA-Seq data, we can only infer that clusters 1–3, which show reduced *Aif1* expression (the gene encoding Iba1) compared with cluster 0 (homeostatic microglia), may account for some of the Iba1-negative microglia at PT D1. However, further analysis of the scRNA-Seq data showed that these microglial clusters did not have higher *Ptprc* expression (the gene encoding CD45). Furthermore, unlike in other studies reporting distinct, injury-induced, proliferation-associated microglial clusters [10], the microglial clusters in our study were close to negative for the proliferation genes *Mki67* and *Top2a* (**Point-by-Point Response Fig. 5a-b**).

Finally, to further describe the SVZ microglial phenotypes, we performed immunohistochemical analysis in microglia-targeting mouse lines. Interestingly, the microglia-targeting *HexB-tdTomato* and *Cx3cr1-CreERT2:Rosa26-tdTomato* mouse lines revealed that a proportion of SVZ microglia expressed Iba1 either at a low level or not at all (**Extended Data Fig. 4b-c** in the revised manuscript). Although the cells did not proliferate (**Extended Data Fig. 5i** in the revised manuscript) and did not express the precursor cell marker IRF8 (**Point-by-Point Response Fig. 2b**), ~40% of SVZ-populating Iba1-RFP+ microglia expressed CD68 (**Extended Data Fig. 4d** in the revised manuscript), a marker expressed by a distinct Iba1- microglia population reported in deep subcortical brain lesions with yet unknown functions [20].

Together, these immunohistochemical analyses confirm the heterogeneous molecular composition of microglia in the distinct SVZ stem cell niche microenvironment. We have added new text to the Results section on **page 7, lines 141-162** in the revised manuscript on further delineation of the SVZ microglial phenotype.

Figure 5. Expression of CD45 and proliferative genes in SVZ microglial clusters. **a**, Dot plot of selected microglia/monocyte-associated genes and proliferative genes expressed by the SVZ microglial clusters. **b**, UMAP plots colored for expression of microglia/monocyte-associated genes and proliferative genes expressed by SVZ microglia.

Q6. Microglia depletion studies: The phenotypic characterisation of microglia to eliminate dying DCX-positive cells is interesting. However, I greatly miss insight into the functional consequences of these events. Do the authors have data to show whether the numbers of newly born neurons were altered by the absence of microglia, was neuroblast migration, proliferation, etc, affected? Did microglia depletion have any impact on the wiring of cortical circuits, did it affect functional recovery, cognition or motor function? Without providing such data the observed changes remain largely descriptive.

We thank the reviewer for raising these important points. We would like to highlight that the focus of our study is to progress towards understanding how cortical stroke might change the interactions between microglia and NSPCs in the SVZ stem cell niche, limiting the neurogenic response for brain repair (**Fig. 1a-c** in the revised manuscript). Our data show that after remote cortical brain injury, the altered niche environment leads to immediate activation of microglia in the SVZ niche and an abnormal neurogenic response, with cell-cycle arrest of neural stem cells and neuroblast death.

As summarized at the start of this response (under “**Major points of the Reviewers**”), we have performed several new experiments to address the pathophysiological significance of the interaction between microglia and neural stem cells in the SVZ after PT.

First, we performed new experiments to investigate whether the increased number of SVZ-derived DCX+ neuroblasts after microglial depletion results in the formation of newborn neurons at the cortical lesion. We

performed several experiments with different PLX5622 feeding regimes to investigate the SVZ-derived neurogenic response (migration and differentiation) of DCX+ neuroblasts towards the cortical lesion site, but failed to observe such migration to the lesion penumbra. Upon cortical brain injury, altered cell- or blood-derived signals from the injury site, such as monocytes, are thought to attract SVZ-derived NSPCs to migrate towards the lesion area, thereby contributing to brain repair [24,25]. Our data show that the PLX5622 diet, as well as leading to successful microglial depletion, also resulted in depletion of blood-derived monocytes in the cortical lesion area. Thus the potential lack of an attracting migratory signal may explain why we did not observe migration of SVZ-derived DCX+ neuroblasts towards the lesion penumbra in mice on the PLX5622 diet (**data not shown**).

Upon cortical stroke, the SVZ stem cell niche environment is drastically changed, with increased permeability of the SVZ vasculature and fibrinogen deposition (**Fig. 2a** in the revised manuscript and [1]), and this also coincides with early SVZ microglial activation (**Fig. 1b, Extended Data Fig. 1b** in the revised manuscript). Fibrinogen has been demonstrated to activate microglia [2,3].

Therefore, we next restored the SVZ niche environment after PT by using anicrod to acutely deplete fibrinogen from the circulation; we then analyzed the SVZ-derived DCX+ neuroblasts in the peri-infarct PT lesion area. Anicrod-treated animals showed a 93% reduction in fibrinogen levels in the brain parenchyma compared with vehicle-treated control animals (**data not shown**). Acute fibrinogen depletion by anicrod resulted in a 60% reduction in the number of activated microglia in the SVZ at 10 days post-injury, compared with control mice (**Extended Data Fig. 5e** in the revised manuscript). Importantly, fibrinogen depletion increased the SVZ-derived DCX+ neurogenic response in the lesion penumbra by ~2-fold (**Fig. 2d** in the revised manuscript).

Next, we analyzed how depleting SVZ microglia affects neuroblast formation and survival. Depletion of microglia with the colony-stimulating factor 1 receptor (CSF1R) inhibitor PLX5622 (starting prior to induction of PT and continuing throughout the experimental period) reduced the DCX+ neuroblast cell death and increased the number of newborn EdU+DCX+ neuroblasts and the overall number of DCX+ cells in the SVZ 7 days after PT (**Fig. 2e** and **Extended Data Fig. 5f-h** in the revised manuscript).

Overall, our data show that microglial depletion increased the formation and survival of newborn neuroblasts in the SVZ and the pharmacological restoration of the niche environment increased the SVZ-derived neurogenic repair after PT. We have added 4 new figure panels (**Fig. 2d-e, Extended Data Fig. 5f-g**) and extra text to the Results (**page 8-9, lines 188-200**) and Discussion (**pages 12-13, lines 269-305**) sections to the revised manuscript, to clarify that activated microglia in the SVZ reduced the SVZ-derived neurogenic response for brain repair. Furthermore, the pharmacological depletion of fibrinogen mentioned above alleviates the stroke-induced SVZ environmental changes and reduces the SVZ ApoE+ microglia (**Fig. 3g** in the revised manuscript), potentially explaining the increased SVZ-derived neurogenic response in fibrinogen-depleted mice after PT (**Fig. 2d** in the revised manuscript). These new data demonstrate the pathophysiological significance of the interaction between SVZ microglia and NSPCs on the neurogenic response after PT.

Q7. Methods: „PT was performed 5 days after the start of the PLX5622 diet and mice were sacrificed 7 days after PT”. Did PLX5622 treatment alter the SVZ in any means prior to stroke or influence infarct size, BBB injury, or astrocyte responses after stroke?

We thank the reviewer for her/his important question. We performed new experiments showing that microglial depletion in uninjured mice does not affect the number of DCX+ neuroblasts in the SVZ stem cell niche prior to stroke (**Extended Data Fig. 5f** in the revised manuscript). Furthermore, new experiments revealed that the depletion of microglia by the PLX5622 diet did slightly alter the infarct size as measured 7 days after PT and reduced GFAP immunoreactivity at the lesion site by ~50%, indicating reduced astrocyte reactivity, potentially due to the reduced inflammation (**Point-by-Point Response Fig. 6a-b**). Overall, the PLX5622 diet does not significantly alter the neuroblast cell number in the SVZ before PT.

Figure 6. The effect of myeloid cell depletion on lesion size and astrocyte reactivity. **a**, Hematoxylin and eosin staining of the brain cortex 7 days after PT in PLX5622-treated mice compared with control-fed mice. The area delineated by the dotted yellow line corresponds to the lesion core. Scale bar, 500 μm . Quantification of the infarct volume ($n=3$ mice). **b**, Immunolabeling for GFAP (green) in the cortex 7 days after PT in PLX5622-treated mice compared with control-fed mice. The area delineated by the dotted yellow line corresponds to the lesion core. Scale bar, 670 μm . Quantification of GFAP-immunoreactivity in a region of interest ($400 \times 200 \mu\text{m}$) in the glial border ($n=4$ mice, control-fed; $n=5$ mice, PLX-fed). All the plots show the mean \pm SEM. * $P<0.05$, ns, not significant. Unpaired Mann–Whitney test.

Q8. The authors identify the ligand–receptor pair *Lrp8–ApoE* in NSPC–microglia cross-communication, which I find important in this context. However, nothing is known about the impact of these interactions on neurogenesis or functional recovery after stroke in this model. Have the authors considered using at least *Lrp8* KO mice or targeting the SVZ compartment with pharmacological tools (e.g. neutralizing antibodies, blocking peptides against *Lrp8*) to study these processes in vivo?

We thank the reviewer for raising this important point. As already mentioned above for Reviewer 1, we are investigating how upon a remote cortical brain injury (cortical PT), the altered SVZ niche environment (with blood-derived fibrinogen deposition) is leading to immediate activation of SVZ microglia and to the abnormal neurogenic response, with cell-cycle arrest of neural stem cells and neuroblast cell death. Overall, our data suggest that the altered niche environment leads to discrete stroke-associated SVZ microglial clusters (cluster 1, cluster 4) that negatively impact SVZ neurogenesis and that the altered cross-communication between microglial subclusters and NSPCs regulates the extent of the innate neurogenic repair response in the SVZ after stroke.

As the *Lrp8* depleted mouse line (JAX #003524) has been described to have several brain developmental defects [4], the conditional knockout mouse line was not available and as intracerebroventricular injection approaches will potentially affect the SVZ microglia – NSPC interaction (Our lab has shown fibrinogen deposition in the SVZ after cortical stab wound injury, please see Pous *et al.* [1], supplementary Figure 1), we decided to pharmacologically restore the niche environment (fibrinogen depletion by anicrod) and to deplete SVZ microglia to investigate the pathophysiological significance of the SVZ microglia – NSPC interaction after PT. Our new data revealed that depletion of fibrinogen reduces SVZ microglial activation and increases the number of DCX+ neuroblasts in the lesion penumbra. Depletion of microglia increases the number of newborn neuroblasts and reduces neuroblast cell death in the SVZ stem cell niche (**Fig. 2d-e**, **Fig. 3g**, **Extended Data Fig. 5e-h** in the revised manuscript).

To further describe the microglia-NSPC ligand-receptor interaction (as already summarized at the start of this response under “Major points of the Reviewers”), we performed new experiments and analyzed the proximity of *Lrp8*+ NSPCs to *ApoE*+ microglia using 3-D IMARIS-processed images. This revealed close contact between *Lrp8*+DCX+ neuroblasts and RFP+ microglia and between *ApoE*+RFP+ microglia and DCX+ neuroblasts in the SVZ of mice 7 days after PT (proximity close to 0 μm) (**Fig. 3e-f** in the revised manuscript). Furthermore, depletion of fibrinogen reduces the number of SVZ *ApoE*+ microglia (**Fig. 3g** in the revised manuscript), suggesting that indeed stroke-induced SVZ microglial cluster regulate the extent of the innate neurogenic repair response in the SVZ by altered microglia-NSPC ligand–receptor interactions after stroke.

An emerging concept is the reciprocal interaction between inflammatory cells and NSPCs in CNS disease [5]. As microglia–NSPC interactions in the healthy CNS and during disease are still poorly defined, our repository of ligand–receptor pairs will help elucidate the biology of microglia – NSPC interactions in future

studies. We added sentences to the Discussion section, **page 15, lines 335-343** in the revised manuscript to emphasize that our repository of ligand–receptor pairs will help elucidate the biology of microglia and NSPCs interaction, which will be addressed in future studies. Furthermore, we changed the title of the manuscript to better focus the manuscript on how the altered SVZ niche environment leads to discrete stroke-associated SVZ microglial clusters that negatively impact SVZ neurogenesis.

Minor point:

- I would update graphics and text to name C cells and B cells „Type C cells” and „Type B cells” to ensure that readers outside the SVZ field understand their relationships and origin.

We thank the reviewer for this comment. We have corrected the graphics and text to name C cells and B cells „Type C cells” and „Type B cells” throughout the revised manuscript.

Reviewer #3:

Comments for the Author:

In this study, the authors aim to address a neurogenic response after stroke that is claimed to be regulated by cross communication between specific types of microglia (MG) and neural stem/progenitor (NSPC) subclusters. Despite the relevance of this topic, in my opinion, the work shown here suffers from a general lack of focus and novelty. The title and the abstract are misleading as the biological question, the interaction between MG and NSPC and its effects on neurogenic responses, is not functionally addressed throughout the whole study. The authors claim to find an interaction between MG and NSPC, but fail to show that this connection regulates the neurogenic response. Essential experiments are missing to draw this conclusion and the data presented is not sufficient to make such a statement. In addition, the quality of the data is worrisome, since in suppl. figure 2 a very high % of mitochondrial genes are shown, suggesting that sequenced cells might be apoptotic/cell death. Along with this, their proof of MG and NSPC interaction is also rather weak and does not go beyond established knowledge in the field. That MG endocytose apoptotic cells are not astonishing, but rather depict what MG are known for. Thus, they are not novel and don't add supporting evidence for their hypothesis regarding the interaction or regulation of the neurogenic response. In sum, the current study consists mainly of repetitions and reproductions of previous studies, for instance the heterogeneity of microglia in the brain and does not add any novel insights or findings into the field of MG and NSPC regulation.

In summary, this study provides neither a new data set nor a new tool and does not present compelling evidence to support its conclusions. It also lacks a functional analysis that would have shed light on cross-communication in the subventricular zone (SVZ) during injury.

We thank the reviewer for her/his criticism. Yet, we argue that our study does indeed add novel insights into the endogenous SVZ repair mechanism after cortical brain injury:

Although it is known that the neurogenic response from the SVZ to repair the brain after a stroke is very limited [25-28], the cause of this limitation is not yet fully understood. In our lab, we are investigating the potential underlying cause of the limited neurogenic response in the SVZ after stroke. In our previous study, we found that distal cortical injury (from photothrombotic ischemia, PT) leads to permeability of the SVZ niche vasculature and massive microenvironmental changes within the SVZ stem cell niche that control NSPC fate, primarily promoting NSPC-derived astrogenesis [1]. Only a few SVZ NSPCs generate young neurons that contribute to the repair process. Here, we study how the changes in SVZ niche vasculature and altered SVZ niche environment affect the interaction between SVZ microglia and NSPCs, and how this interaction potentially impacts the neurogenic response after stroke. We present here the following new insights:

i In this study, we show that the SVZ is a vulnerable region, transmitting the increased permeability of the SVZ vasculature and the changed niche microenvironment (e.g., fibrinogen deposition) upon distant cortical injury into early SVZ niche microglial activation (**Fig. 1b, Fig. 2a, Extended Data Fig. 1b** in the revised manuscript). In line, fibrinogen depletion by the pharmacological reagent anicrod resulted in a 60% reduction in activated microglia in the SVZ at 10 days post-injury, compared with control mice (**Extended Data Fig. 5e** in the revised manuscript).

ii As well as the increased number of activated type B cells (**Extended Data Fig. 3b-c** in the revised manuscript), as expected, our data highlight that the altered niche environment after remote cortical brain injury leads to an abnormal neurogenic response, with cell-cycle arrest of neural stem cells and neuroblast cell death within the SVZ (**Fig. 1h-j, Extended Data Fig. 3g-h** in the revised manuscript).

iii We highlight that discrete stroke-associated SVZ microglial clusters reduce the neurogenic response. Microglia in the germinal niche are known to have a distinct phenotype supporting neurogenesis under both homeostasis [29,30] and during the reparative processes after stroke [31]. However, we found that this homeostatic, pro-neurogenic microglial phenotype switched drastically in the first days after cortical stroke to discrete stroke-associated SVZ microglial states that negatively impact the neurogenic response (**Fig. 1k-m, Figs. 2b-c, 2f-g, Extended Data Fig. 4, Extended Data Fig. 5a-d** in the revised manuscript).

iv As already summarized at the start of this response (under “**major points of the Reviewers**”), we have performed several new experiments to address the pathophysiological significance of SVZ microglia–NSPC interactions after PT. Microglial depletion increased the formation and survival of newborn neuroblasts in the SVZ and the pharmacological restoration of the niche environment increased the SVZ-derived neurogenic repair after PT. Depletion of fibrinogen reduces SVZ microglial activation, reduces the number of SVZ ApoE+ microglia, and increases the number of DCX+ neuroblasts in the lesion penumbra. Depletion of microglia increases the number of newborn neuroblasts and reduces neuroblast cell death in the SVZ stem cell niche (**Fig. 2d-e, Fig. 3g, Extended Data Fig. 5e-h** in the revised manuscript). Overall, our data suggest that the altered niche environment leads to distinct SVZ microglia clusters that negatively impact the innate neurogenic response early after stroke.

v Microglial heterogeneity occurs in a regional- and context-dependent manner owing to their high responsiveness to diverse environmental conditions [9]. Nevertheless, the heterogeneity of microglia in the neurogenic SVZ niche during reparative processes after stroke is only poorly described. To the best of our knowledge, our study reports for the first time the different SVZ microglial states under homeostatic conditions and after PT, assessed by scRNA-Seq analysis.

vi We also propose for the first time a repository of SVZ microglia–NSPC ligand–receptor pairs that are altered chronologically after stroke (**Fig. 3a** in the revised manuscript). Our data suggest that this changed cell communication arises from the altered SVZ niche environment (**Fig. 3g** in the revised manuscript). Therefore, our study not only sheds light on the distinctive and specific transcriptional status of microglia in the neurogenic niche under homeostasis and with CNS damage, but also proposes new therapeutic possibilities related to modulating the neurogenic environment and/or cell–cell interactions to promote endogenous regeneration.

We believe we have significantly revised and improved the focus and clarity of our manuscript. We changed the title of the manuscript to better focus the manuscript on how the altered SVZ niche environment leads to discrete stroke-associated SVZ microglial clusters that negatively impact SVZ neurogenesis. We added new references and thoughts to the discussion to highlight the impact and novelty of our results, and that our study represents a significant advance towards harnessing endogenous NSPC therapy in CNS diseases, including stroke but also potentially other CNS diseases as well.

General remarks on the manuscript:

- Line 65: provide an explanation of what YFP labels for and explain the mouse models.

We thank the reviewer for this critical observation. We now indicate that YFP labels SVZ NSPCs targeted by the *Nestin-CreER^{T2}:R26-yfp* transgenic mouse line [21] and have added this information to **page 4, lines 81-84** in the revised manuscript. Furthermore, we introduced the mouse model used for cortical ischemic stroke—photothrombotic ischemia, PT—on **page 3, lines 63 and 68**, of the revised manuscript.

Figures and Analysis:

The text and figure captions mention that scRNAseq data has been visualized using Diffusion maps, but the figures show UMAP coordinate systems while the methods don't mention diffusion maps. Please clarify what method was used where.

We thank the reviewer for this insightful observation and we regret the mistake. The SVZ microglia and NSPC clusters were explored and visualized with UMAP, not diffusion maps. All corresponding text and figures have been adjusted accordingly in the revised manuscript.

Figure 1F-G: the quantification should be done as the ratio of apoptotic cells DCX-Apoptag positive vs all NBs or Mash1 cells. As it is, the single fact that there is a higher NB abundance at 7 dpi could show the same result, without altering NB probability to undergo apoptosis.

We thank the reviewer for this insightful comment and we have revised the figures and figure legends accordingly (**Fig. 1j** and **Extended Data Fig. 3h** in the revised manuscript).

Figure 2A: the reviewer cannot see the continuum between cluster 0 and 1 and it is not clear why two clusters (2,3) are not present.

We thank the reviewer for this important criticism. We have now removed this analysis, because our new trajectory analysis including all microglial clusters resulted in no further clarification of the origin of phagocytic microglia at day 7 after PT. The revised manuscript excludes the trajectory analysis of the SVZ microglia (**Fig. 2a** in the original manuscript).

Figure 3: the MG-SVZ NSCs interaction based on immunohistochemistry is too weak to draw the cross-connection conclusion.

We want to thank the reviewer for this important point. We have further characterized the SVZ microglia–NSPC ligand–receptor pairs after PT, as outlined above, and also analyzed the proximity of Lrp8+ NSPCs to ApoE+ microglia using 3-D IMARIS-processed images. This revealed close contact between Lrp8+DCX+ neuroblasts and RFP+ microglia as well as between ApoE+RFP+ microglia and DCX+ neuroblasts in the SVZ of mice 7 days after PT (proximity close to 0 μm) (**Fig. 3e-f** in the revised manuscript).

In the revised manuscript we show that the interaction between SVZ microglia and neuroblasts reduced the neurogenic response after PT. We propose that the interaction between Lrp8+ NSPCs and ApoE+ microglia could be a potential target for future studies. Indeed, our results show a drastically changed SVZ microenvironment with increased fibrinogen deposition after PT (**Fig. 2a** in the revised manuscript); furthermore, pharmacological depletion of fibrinogen alleviated the stroke-induced change in the SVZ environment, reduced the number of activated SVZ microglia, and reduced the number of SVZ ApoE+ microglia (**Fig. 3g, Extended Data Fig. 5e** in the revised manuscript), potentially explaining the increased neurogenic response in fibrinogen-depleted mice after PT (**Fig. 2d** in the revised manuscript).

The reported size of scale bars in microscopy images seems unrealistic as they are in picometers and the labels of the scales must be separated clearer.

We have carefully revisited and corrected all scale bars in the revised manuscript.

Figure 4: The reporting summary states that test statistics for null hypothesis testing have been reported, but no test statistics are provided.

We have carefully revised the statistical analyses and corrected the reporting summary accordingly.

Further on extended Figures:

Figure 1A: remove results citation from the introduction.

The citation of Extended Figure 1a (**Fig. 1c** in the revised manuscript) has been removed from the Introduction in the revised manuscript.

Figure 1b: misspelling of dissociation.

The misspelling in Extended Figure 1A (now **Fig. 1c** in the revised manuscript) has been addressed.

Figure 2: the % of mitochondrial genes is very very high, suggesting that sequenced cells might be apoptotic/cell death.

We agree with the reviewer that the mitochondrial content of the data is high compared to what is typically found in scRNA-Seq datasets, especially compared to data from droplet-based methods [32]. However, we believe our data are of sufficiently good quality to support the claims in the manuscript for several reasons:

i) We found that the mitochondrial content was not a distinguishing feature of any of the NSPC or microglial clusters, nor were mitochondrial genes among the highest-ranked DEGs except for activated type B cells. Researchers typically exclude data from scRNA-Seq analyses if there are low-quality cells, distinct clusters with low numbers of unique genes, low numbers of UMIs, high mitochondrial content, or a combination of all of these. However, none of these issues were detected in our scRNA-Seq dataset.

ii) We performed an integrated analysis by combining our data with the data published by Zywitza *et al.* 2018 [33], in which the authors similarly studied cells of the SVZ. This analysis identified shared cell types between the two studies despite difference in the mitochondrial content (**Point-by-Point Response Fig. 7a**).

iii) Our use of a lenient mitochondrial threshold (in our case, 25%) is not unique among scRNA analyses; similar thresholds have been used in previous studies analyzing neural stem cell niches or other CNS tissues [34,35]. Furthermore, it has been demonstrated that mitochondrial content varies in a cell-type-dependent manner and that CNS tissues rank among the highest average mitochondrial content [36]. In agreement with these findings, when we examined the mitochondrial content of our data by cell-type, we observed that NSPCs displayed a higher average mitochondrial content than microglia (**Point-by-Point Response Fig. 7b**).

Figure 7. The high mitochondrial content does not interfere with the validity of the scRNA-Seq dataset. a, t-SNE plot of microglia cells with the time point of origin compared with Zywitza’s study [33]. b, Violin plots of mitochondrial genes enriched in different cell populations.

Figure 3: given the small size and low resolution of the images, it is hard for the reader to observe double positive cells. In addition, the signal seems overexposed.

We thank the reviewer for his/her insightful comment. We have carefully revised the images by increasing the size, adding an asterisk to double-positive cells and added orthogonal views to the images (**Fig. 1i-j** in the revised manuscript).

Figure 4: the authors did not consider CreERT2 inefficient recombination. And the HexB and Cx3cr1 strategy does not add on microglial heterogeneity when already scRNA-seq data is providing this information.

We thank the reviewer for this comment. Our scRNA-Seq analysis comparing different time points after PT and uninjured mice revealed different SVZ microglial activation states. Immunolabeling for purinergic signaling of SVZ microglia to characterize their functionality has so far been conducted in healthy mice [30].

We further delineated the SVZ microglia phenotype by immunohistochemical labeling and we only describe the Iba1 (ionized calcium-binding adaptor molecule 1) expression in targeted RFP+ microglia in the revised manuscript. Therefore, an inefficient recombination efficiency of the microglia targeting mouse lines is not applicable. We carefully re-examined the SVZ microglial phenotype and fine-tuned the Iba1 expression threshold to strictly distinguish SVZ microglia with a low level of Iba1 expression from microglia with no detectable Iba1 expression (Methods section **page 30, lines 698-701** in the revised manuscript).

We describe heterogenous microglia activation states in the SVZ stem cell niche under homeostatic conditions and after PT. Our revised data confirm that immunolabeling for Iba1 in uninjured HexB-tdTomato transgenic mice resulted in ~100% co-labeling of cortical microglia (**Extended Data Fig. 4b** in the revised manuscript). By contrast, ~40% of HexB+ cells (red) in the SVZ either had a low level of Iba1 expression or were negative for Iba1 (green) (**Extended Data Fig. 4b** in the revised manuscript). In line with this, immunolabeling for Iba1 in Cx3cr1-CreERT2:R26-tdTomato mice at day 1 and day 7 after PT and in uninjured mice revealed that ~15%–20% of RFP+ SVZ microglia either had low expression levels or were negative for Iba1 (green) (**Extended Data Fig. 4c** in the revised manuscript). Finally, ~40% of SVZ-

populating Iba1-RFP+ microglia expressed CD68 (**Extended Data Fig. 4d** in the revised manuscript), a marker expressed by a distinct Iba1- microglia population reported in deep subcortical brain lesions with yet unknown functions [20]. Overall, this immunohistochemical data extend our scRNA-Seq data on the SVZ microglia heterogeneity on the protein level and add new information on novel Iba1- microglia cells in the SVZ stem cell niche.

References

1. Pous, L.; Deshpande, S.S.; Nath, S.; Mezey, S.; Malik, S.C.; Schildge, S.; Bohrer, C.; Topp, K.; Pfeifer, D.; Fernandez-Klett, F.; et al. Fibrinogen induces neural stem cell differentiation into astrocytes in the subventricular zone via BMP signaling. *Nat Commun* **2020**, *11*, 630, doi:10.1038/s41467-020-14466-y.
2. Adams, R.A.; Bauer, J.; Flick, M.J.; Sikorski, S.L.; Nuriel, T.; Lassmann, H.; Degen, J.L.; Akassoglou, K. The fibrin-derived gamma377-395 peptide inhibits microglia activation and suppresses relapsing paralysis in central nervous system autoimmune disease. *J Exp Med* **2007**, *204*, 571-582.
3. Petersen, M.A.; Ryu, J.K.; Akassoglou, K. Fibrinogen in neurological diseases: mechanisms, imaging and therapeutics. *Nat Rev Neurosci* **2018**, *19*, 283-301, doi:10.1038/nrn.2018.13.
4. Trommsdorff, M.; Gotthardt, M.; Hiesberger, T.; Shelton, J.; Stockinger, W.; Nimpf, J.; Hammer, R.E.; Richardson, J.A.; Herz, J. Reeler/Disabled-like disruption of neuronal migration in knockout mice lacking the VLDL receptor and ApoE receptor 2. *Cell* **1999**, *97*, 689-701, doi:10.1016/s0092-8674(00)80782-5.
5. Peruzzotti-Jametti, L.; Bernstock, J.D.; Vicario, N.; Costa, A.S.H.; Kwok, C.K.; Leonardi, T.; Booty, L.M.; Bucci, I.; Balzarotti, B.; Volpe, G.; et al. Macrophage-Derived Extracellular Succinate Licenses Neural Stem Cells to Suppress Chronic Neuroinflammation. *Cell Stem Cell* **2018**, *22*, 355-368 e313, doi:10.1016/j.stem.2018.01.020.
6. Codega, P.; Silva-Vargas, V.; Paul, A.; Maldonado-Soto, A.R.; Deleo, A.M.; Pastrana, E.; Doetsch, F. Prospective identification and purification of quiescent adult neural stem cells from their in vivo niche. *Neuron* **2014**, *82*, 545-559.
7. Pastrana, E.; Cheng, L.C.; Doetsch, F. Simultaneous prospective purification of adult subventricular zone neural stem cells and their progeny. *Proc Natl Acad Sci U S A* **2009**, *106*, 6387-6392, doi:10.1073/pnas.0810407106.
8. Jordao, M.J.C.; Sankowski, R.; Brendecke, S.M.; Sagar, Locatelli, G.; Tai, Y.H.; Tay, T.L.; Schramm, E.; Armbruster, S.; Hagemeyer, N.; et al. Single-cell profiling identifies myeloid cell subsets with distinct fates during neuroinflammation. *Science* **2019**, *363*, doi:10.1126/science.aat7554.
9. Masuda, T.; Sankowski, R.; Staszewski, O.; Prinz, M. Microglia Heterogeneity in the Single-Cell Era. *Cell Rep* **2020**, *30*, 1271-1281, doi:10.1016/j.celrep.2020.01.010.
10. Sankowski, R.; Bottcher, C.; Masuda, T.; Geirsdottir, L.; Sagar, Sindram, E.; Sereidenina, T.; Muhs, A.; Scheiwe, C.; Shah, M.J.; et al. Mapping microglia states in the human brain through the integration of high-dimensional techniques. *Nat Neurosci* **2019**, *22*, 2098-2110, doi:10.1038/s41593-019-0532-y.
11. Masuda, T.; Sankowski, R.; Staszewski, O.; Bottcher, C.; Amann, L.; Sagar, Scheiwe, C.; Nessler, S.; Kunz, P.; van Loo, G.; et al. Spatial and temporal heterogeneity of mouse and human microglia at single-cell resolution. *Nature* **2019**, *566*, 388-392, doi:10.1038/s41586-019-0924-x.
12. Keren-Shaul, H.; Spinrad, A.; Weiner, A.; Matcovitch-Natan, O.; Dvir-Szternfeld, R.; Ulland, T.K.; David, E.; Baruch, K.; Lara-Astaiso, D.; Toth, B.; et al. A Unique Microglia Type Associated with Restricting Development of Alzheimer's Disease. *Cell* **2017**, *169*, 1276-1290 e1217, doi:10.1016/j.cell.2017.05.018.
13. Krasemann, S.; Madore, C.; Cialic, R.; Baufeld, C.; Calcagno, N.; El Fatimy, R.; Beckers, L.; O'Loughlin, E.; Xu, Y.; Fanek, Z.; et al. The TREM2-APOE Pathway Drives the Transcriptional Phenotype of Dysfunctional Microglia in Neurodegenerative Diseases. *Immunity* **2017**, *47*, 566-581 e569, doi:10.1016/j.immuni.2017.08.008.
14. Hammond, T.R.; Dufort, C.; Dissing-Olesen, L.; Giera, S.; Young, A.; Wysoker, A.; Walker, A.J.; Gergits, F.; Segel, M.; Nemes, J.; et al. Single-Cell RNA Sequencing of Microglia throughout the Mouse Lifespan and in the Injured Brain Reveals Complex Cell-State Changes. *Immunity* **2019**, *50*, 253-271 e256, doi:10.1016/j.immuni.2018.11.004.

15. Ransohoff, R.M. A polarizing question: do M1 and M2 microglia exist? *Nat Neurosci* **2016**, *19*, 987-991, doi:10.1038/nn.4338.
16. Paolicelli, R.C.; Sierra, A.; Stevens, B.; Tremblay, M.E.; Aguzzi, A.; Ajami, B.; Amit, I.; Audinat, E.; Bechmann, I.; Bennett, M.; et al. Microglia states and nomenclature: A field at its crossroads. *Neuron* **2022**, *110*, 3458-3483, doi:10.1016/j.neuron.2022.10.020.
17. Murray, P.J. Macrophage Polarization. *Annu Rev Physiol* **2017**, *79*, 541-566, doi:10.1146/annurev-physiol-022516-034339.
18. Kierdorf, K.; Erny, D.; Goldmann, T.; Sander, V.; Schulz, C.; Perdiguero, E.G.; Wieghofer, P.; Heinrich, A.; Riemke, P.; Holscher, C.; et al. Microglia emerge from erythromyeloid precursors via Pu.1- and Irf8-dependent pathways. *Nat Neurosci* **2013**, *16*, 273-280, doi:10.1038/nn.3318.
19. Lewis, S.M.; Williams, A.; Eisenbarth, S.C. Structure and function of the immune system in the spleen. *Science immunology* **2019**, *4*, doi:10.1126/sciimmunol.aau6085.
20. Waller, R.; Baxter, L.; Fillingham, D.J.; Coelho, S.; Pozo, J.M.; Mozumder, M.; Frangi, A.F.; Ince, P.G.; Simpson, J.E.; Highley, J.R. Iba-1-/CD68+ microglia are a prominent feature of age-associated deep subcortical white matter lesions. *PLoS One* **2019**, *14*, e0210888, doi:10.1371/journal.pone.0210888.
21. Lagace, D.C.; Whitman, M.C.; Noonan, M.A.; Ables, J.L.; DeCarolis, N.A.; Arguello, A.A.; Donovan, M.H.; Fischer, S.J.; Farnbauch, L.A.; Beech, R.D.; et al. Dynamic contribution of nestin-expressing stem cells to adult neurogenesis. *J Neurosci* **2007**, *27*, 12623-12629.
22. Jahn, H.M.; Kasakow, C.V.; Helfer, A.; Michely, J.; Verkhatsky, A.; Maurer, H.H.; Scheller, A.; Kirchhoff, F. Refined protocols of tamoxifen injection for inducible DNA recombination in mouse astroglia. *Sci Rep* **2018**, *8*, 5913, doi:10.1038/s41598-018-24085-9.
23. Sun, M.Y.; Yetman, M.J.; Lee, T.C.; Chen, Y.; Jankowsky, J.L. Specificity and efficiency of reporter expression in adult neural progenitors vary substantially among nestin-CreER(T2) lines. *J Comp Neurol* **2014**, *522*, 1191-1208, doi:10.1002/cne.23497.
24. Ohab, J.J.; Fleming, S.; Blesch, A.; Carmichael, S.T. A neurovascular niche for neurogenesis after stroke. *J Neurosci* **2006**, *26*, 13007-13016.
25. Thored, P.; Arvidsson, A.; Cacci, E.; Ahlenius, H.; Kallur, T.; Darsalia, V.; Ekdahl, C.T.; Kokaia, Z.; Lindvall, O. Persistent production of neurons from adult brain stem cells during recovery after stroke. *Stem Cells* **2006**, *24*, 739-747.
26. Arvidsson, A.; Collin, T.; Kirik, D.; Kokaia, Z.; Lindvall, O. Neuronal replacement from endogenous precursors in the adult brain after stroke. *Nat Med* **2002**, *8*, 963-970.
27. Liang, H.; Zhao, H.; Gleichman, A.; Machnicki, M.; Telang, S.; Tang, S.; Rshtouni, M.; Ruddell, J.; Carmichael, S.T. Region-specific and activity-dependent regulation of SVZ neurogenesis and recovery after stroke. *Proc Natl Acad Sci U S A* **2019**, *116*, 13621-13630, doi:10.1073/pnas.1811825116.
28. Williamson, M.R.; Le, S.P.; Franzen, R.L.; Donlan, N.A.; Rosow, J.L.; Nicot-Cartsonis, M.S.; Cervantes, A.; Deneen, B.; Dunn, A.K.; Jones, T.A.; et al. Subventricular zone cytotogenesis provides trophic support for neural repair in a mouse model of stroke. *Nat Commun* **2023**, *14*, 6341, doi:10.1038/s41467-023-42138-0.
29. Shigemoto-Mogami, Y.; Hoshikawa, K.; Goldman, J.E.; Sekino, Y.; Sato, K. Microglia enhance neurogenesis and oligodendrogenesis in the early postnatal subventricular zone. *J Neurosci* **2014**, *34*, 2231-2243, doi:10.1523/JNEUROSCI.1619-13.2014.
30. Ribeiro Xavier, A.L.; Kress, B.T.; Goldman, S.A.; Lacerda de Menezes, J.R.; Nedergaard, M. A Distinct Population of Microglia Supports Adult Neurogenesis in the Subventricular Zone. *J Neurosci* **2015**, *35*, 11848-11861, doi:10.1523/JNEUROSCI.1217-15.2015.
31. Thored, P.; Wood, J.; Arvidsson, A.; Cammenga, J.; Kokaia, Z.; Lindvall, O. Long-term neuroblast migration along blood vessels in an area with transient angiogenesis and increased vascularization after stroke. *Stroke* **2007**, *38*, 3032-3039.
32. Ding, J.; Adiconis, X.; Simmons, S.K.; Kowalczyk, M.S.; Hession, C.C.; Marjanovic, N.D.; Hughes, T.K.; Wadsworth, M.H.; Burks, T.; Nguyen, L.T.; et al. Systematic comparison of single-cell and single-nucleus RNA-sequencing methods. *Nat Biotechnol* **2020**, *38*, 737-746, doi:10.1038/s41587-020-0465-8.
33. Zywitza, V.; Misios, A.; Bunatyan, L.; Willnow, T.E.; Rajewsky, N. Single-Cell Transcriptomics Characterizes Cell Types in the Subventricular Zone and Uncovers Molecular Defects Impairing Adult Neurogenesis. *Cell Rep* **2018**, *25*, 2457-2469 e2458, doi:10.1016/j.celrep.2018.11.003.

34. Magnusson, J.P.; Zamboni, M.; Santopolo, G.; Mold, J.E.; Barrientos-Somarribas, M.; Talavera-Lopez, C.; Andersson, B.; Frisen, J. Activation of a neural stem cell transcriptional program in parenchymal astrocytes. *eLife* **2020**, *9*, doi:10.7554/eLife.59733.
35. Mendiola, A.S.; Ryu, J.K.; Bardehle, S.; Meyer-Franke, A.; Ang, K.K.; Wilson, C.; Baeten, K.M.; Hanspers, K.; Merlini, M.; Thomas, S.; et al. Transcriptional profiling and therapeutic targeting of oxidative stress in neuroinflammation. *Nat Immunol* **2020**, *21*, 513-524, doi:10.1038/s41590-020-0654-0.
36. Osorio, D.; Cai, J.J. Systematic determination of the mitochondrial proportion in human and mice tissues for single-cell RNA-sequencing data quality control. *Bioinformatics* **2021**, *37*, 963-967, doi:10.1093/bioinformatics/btaa751.

REVIEWERS' COMMENTS

Reviewer #1 (Remarks to the Author):

The manuscript entitled, “Interaction between subventricular zone microglia and neural stem cells impacts the 2 neurogenic response after stroke” by Nath and colleagues is a revised manuscript that seeks to provide mechanistic insight into why neurogenic responses of the SVZ to stroke is limited. The authors of the manuscript combine immunohistochemical and genetic labeling to perform a combination of imaging and single cell RNA sequencing experiments. In my humble opinion, the authors have answered nearly all of the requests of the reviewers. Generally speaking, the data is of high quality and the single cell RNA sequencing is similar several in the field (Troy+ brain stem cells cycle through quiescence and regulate their number by sensing niche occupancy) and ours. The rigor and reproducibility appears high despite the comment of one reviewer. There were concerns of mitochondrial transcripts in the sequencing, but are present in nearly all data. As the authors now indicate, this does not interfere with the interpretation. Most single cell and single nuclei sequencing studies pick up mitochondrial transcripts and these differ based on the cell types and conditions and therefore I am less concerned about the quality of the sequencing. Moreover, the addition of confirmatory immunohistochemistry demonstrating increased NSPC EdU/Sox2/GFAP supports the authors conclusion that stroke increased NSPC activation. The addition of microglia labeling in transgenic mice and staining has further strengthened this idea. It would be nice if extended figures 4/5 could be incorporated into the main text but understand the difficulty of doing this. The extensive description of microglia changes including the sphericity and process (dendritic) length and the incorporation of the UMI counts in microglia of Ascl1 and DCX further supports the conclusion.

In general, the manuscript has improved further and is of good quality. The timeliness of publication is important since the original submission was nearly a year ago and publication of single cell sequencing of similar studies continue. The use of PLX5622 and anicrod provide some additional functional relevance for the interactions and effect on neurogenesis, though these loosely change the focus of the paper and make it more complex, it does support the authors original goals to understand the mechanisms why SVZ neurogenesis is incomplete during stroke.

Minor-

I would support adding the choroid plexus data since it appears that this data is complementary and if it is not included at least in supplemental, it may go unpublished. This would also help improve the perceived novelty of the manuscript. Nevertheless, this is up to the authors discretion.

Line 183-187: In describing Extended Data Fig. 5i-j, you do not adequately describe which mouse you are talking about and mention RFP+ microglia. But the last mention in line 179 is of a mouse that should have GFP microglia and RFP NSPCs and progeny. In the Figure 5d it is labeled as Cx3cr1-CREERT2-tdTomato. Please add this to the text to improve clarity.

Line 1089: there is an extra parenthesis

Data availability: The GEO data number currently reads XXXX. Replace.

Code availability: Novel code should be deposited under github or similar.

Reviewer #2 (Remarks to the Author):

The authors responded to my queries appropriately. One remaining issue to address: Figure 6 data in the response file (Fig.6 A and B panels) should be added to the manuscript as it is needed for the interpretation of the complex data sets presented.

Reviewer #3 (Remarks to the Author):

In the revised version of the manuscript the authors have addressed the previous concerns with enough detail. The incorporated experiments in which authors deplete microglia or fibrinogen highlight the functional role of the interactions between microglia and NSPCs that the authors describe.

Nonetheless, I consider important that that authors address the following comments:

- I appreciate that the authors integrate their data with Zywitza et al. 2018 and show the percentage of mitochondrial genes in their own data in the Figure 7 from the Point-by-point Response. This figure should also be included in the revised manuscript. It would also be helpful to include the enrichment of mitochondrial genes in the reference data in the panel B, alongside the violin plots of the data generated for this study.

- I also recommend the use of one single nomenclature for the diverse cell states in the SVZ niche, out of consistency. If authors choose the identification by type A, B and C cells, neuroblasts should be referred to as "type-A cells".

- In the discussion, authors propose that, upon injury, an increased NSPC proliferation is recorded, together with an arrest of cell cycle. These statements seem contrasting and would benefit from further clarification.

Authors put forward an interesting hypothesis and shed light into the active role of microglia during SVZ activation upon injury.

Point-by-Point Response

We thank the editor and all three reviewers for their insightful and constructive comments and suggestions, further improving and shaping our manuscript.

As suggested by the Reviewers 1, 2 and 3, we now include the **Point-by-Point Response Figs. 3, 6 and 7** into the **Supplementary Figures (Supplementary Figs. 2e-f, 5g, and 6d-e)** in the revised manuscript. Furthermore, we addressed all minor comments raised by Reviewers 1 and 3. A detailed point-by-point reply follows below and all changes in the revised manuscript are in blue.

Reviewer #1:

Comments for the Author:

The manuscript entitled, “Interaction between subventricular zone microglia and neural stem cells impacts the neurogenic response after stroke” by Nath and colleagues is a revised manuscript that seeks to provide mechanistic insight into why neurogenic responses of the SVZ to stroke is limited. The authors of the manuscript combine immunohistochemical and genetic labeling to perform a combination of imaging and single cell RNA sequencing experiments. In my humble opinion, the authors have answered nearly all of the requests of the reviewers. Generally speaking, the data is of high quality and the single cell RNA sequencing is similar several in the field (Troy+ brain stem cells cycle through quiescence and regulate their number by sensing niche occupancy) and ours. The rigor and reproducibility appears high despite the comment of one reviewer. There were concerns of mitochondrial transcripts in the sequencing, but are present in nearly all data. As the authors now indicate, this does not interfere with the interpretation. Most single cell and single nuclei sequencing studies pick up mitochondrial transcripts and these differ based on the cell types and conditions and therefore I am less concerned about the quality of the sequencing. Moreover, the addition of confirmatory immunohistochemistry demonstrating increased NSPC EdU/Sox2/GFAP supports the authors conclusion that stroke increased NSPC activation. The addition of microglia labeling in transgenic mice and staining has further strengthened this idea. It would be nice if extended figures 4/5 could be incorporated into the main text but understand the difficulty of doing this. The extensive description of microglia changes including the sphericity and process (dendritic) length and the incorporation of the UMI counts in microglia of Ascl1 and DCX further supports the conclusion.

In general, the manuscript has improved further and is of good quality. The timeliness of publication is important since the original submission was nearly a year ago and publication of single cell sequencing of similar studies continue. The use of PLX5622 and ancrod provide some additional functional relevance for the interactions and effect on neurogenesis, though these loosely change the focus of the paper and make it more complex, it does support the authors original goals to understand the mechanisms why SVZ neurogenesis is incomplete during stroke.

We thank the reviewer for her/his positive feedback on our manuscript and the constructive comments and suggestions during the revision process, improving and shaping our manuscript.

Minor Comments

I would support adding the choroid plexus data since it appears that this data is complementary and if it is not included at least in supplemental, it may go unpublished. This would also help improve the perceived novelty of the manuscript. Nevertheless, this is up to the authors discretion.

We thank the reviewer for this insightful suggestion. We have added the **Point-by-Point Response Fig. 3** to the **Supplementary Fig. 5g** in the revised manuscript and updated the corresponding figure legend.

Line 183-187: In describing Extended Data Fig. 5i-j, you do not adequately describe which mouse you are talking about and mention RFP+ microglia. But the last mention in line 179 is of a mouse that should have GFP microglia and RFP NSPCs and progeny. In the Figure 5d it is labeled as Cx3cr1-CREERT2-tdTomato. Please add this to the text to improve clarity.

We added the information on the used mouse line to the text on **page 9, lines 191-192** in the revised manuscript.

Line 1089: there is an extra parenthesis

We thank the reviewer for this insightful observation. However, the extra parenthesis is correctly placed (Figure legend of **Supplementary Fig. 6f** in the revised manuscript).

Data availability: The GEO data number currently reads XXXX. Replace. Code availability: Novel code should be deposited under github or similar.

We now provide the data and code availability in the revised manuscript.

Reviewer #2:

Comments for the Author:

The authors responded to my queries appropriately. One remaining issue to address: Figure 6 data in the response file (Fig.6 A and B panels) should be added to the manuscript as it is needed for the interpretation of the complex data sets presented.

We would like to thank the reviewer for her/his constructive comments and suggestions, improving our manuscript. We have added the **Point-by-Point Response Fig. 6** to the **Supplementary Fig. 6d-e** in the revised manuscript and updated the corresponding figure legend.

Reviewer #3:

Comments for the Author:

In the revised version of the manuscript the authors have addressed the previous concerns with enough detail. The incorporated experiments in which authors deplete microglia or fibrinogen highlight the functional role of the interactions between microglia and NSPCs that the authors describe. Nonetheless, I consider important that that authors address the following comments: I appreciate that the authors integrate their data with Zywitza et al. 2018 and show the percentage of mitochondrial genes in their own data in the Figure 7 from the Point-by-point Response. This figure should also be included in the revised manuscript. It would also be helpful to include the enrichment of mitochondrial genes in the reference data in the panel B, alongside the violin plots of the data generated for this study.

We would like to thank the reviewer for her/his insightful comments and suggestions, further improving our manuscript. We added the **Point-by-Point Response Fig. 7** to the **Supplementary Fig. 2e-f** in the revised manuscript and updated the corresponding figure legend. We did not integrate the percentage of mitochondrial genes from our study with the reference data in a new figure panel, as they were performed under different sequencing platforms. Instead, we added text to the method section, **page 20, lines 459-461** in the revised manuscript, to further state the accuracy of our sequencing data.

I also recommend the use of one single nomenclature for the diverse cell states in the SVZ niche, out of consistency. If authors choose the identification by type A, B and C cells, neuroblasts should be referred to as “type-A cells”.

We would like to thank the reviewer for her/his insightful suggestion. We introduce the term ‘type A cell’ as doublecortin-positive (DCX+) newborn neuroblast on **page 4, line 74** in the revised manuscript. However, we keep using the term ‘neuroblast’ in our manuscript, as this term is widely used and thus, better perceived by the broad readers of the journal, and represents better the SVZ-derived neurogenic response, the major focus of our study.

In the discussion, authors propose that, upon injury, an increased NSPC proliferation is recorded, together with an arrest of cell cycle. These statements seem contrasting and would benefit from further clarification.

We thank the reviewer for this insightful comment. We now added text to the discussion on **page 12, line 283** in the revised manuscript to clarify that, while NSPCs revealed an increased proliferation, only a fraction of type C cells undergo cell cycle arrest upon environmental changes in the SVZ niche after PT.

Authors put forward an interesting hypothesis and shed light into the active role of microglia during SVZ activation upon injury.